# Influence of individual models and studies on quantitative mitigation findings in the IPCC Sixth Assessment Report

**Ida Sognnaes** ✉ **& Glen P. Peters**

Quantitative mitigation findings based on emissions scenarios submitted to the Intergovernmental Panel on Climate Change (IPCC) play an authoritative role in climate policy and decision making. We analyse the impact of the uneven representation of models and modelling studies in the IPCC Sixth Assessment Report (AR6) on statistical values that are used to present quantitative mitigation findings. We find that several key AR6 findings are influenced considerably by the model with the most scenarios, including emissions reductions by 2030 and the decline in fossil fuels consistent with 1.5 °C, and we find that the year of net-zero greenhouse gas emissions is influenced considerably by both the model and the study with the most scenarios. We find that weighting by model- or study does not provide a straightforward solution and discuss three issues related to the use of database statistics to present emissions scenarios findings. Informed by the purpose of the IPCC and the kinds of insights that can be obtained from emissions scenarios, we suggest improvements to the assessment of emissions scenarios.

Quantitative findings in the Intergovernmental Panel on Climate Change (IPCC) Working Group III (WGIII) reports on climate mitigation rely strongly on scenarios in the IPCC scenarios databases[1–6]. Descriptive database statistics, including median values and interquartile or 5th-95th percentile ranges, are used to report key findings, including emissions reductions over time, the year of net-zero emissions, and the reductions in fossil fuels consistent with different temperature targets[2]. These findings play authoritative roles and have been used to inform climate negotiations and climate policy at international and national levels[7]. Increasingly, they are also used by other actors, including local governments, private companies, banks, and financial regulators to inform net-zero strategies[8], evaluate alignment with climate targets[8], and to assess and disclose climate-related financial risks[8–10].

The emissions scenarios in the IPCC scenarios databases are generated almost exclusively by process-based Integrated Assessment Models (IAMs). The collection is based on the voluntary submission of scenarios that have been published in individual modelling studies or in multi-model studies[11,12]. Because some modelling groups publish and submit more scenarios, the number of scenarios from different models

and studies in the database is not even[12,13]. For this reason, the IPCC scenarios databases are referred to as "ensembles of opportunity"[14]. In the Sixth Assessment Report (AR6), the four models with the most scenarios are responsible for two-thirds of the scenarios that passed vetting and received a climate assessment, and the study with the most scenarios is responsible for almost half of the scenarios (Fig. 1).

Because scenario outcomes depend on model and study assumptions[15–18], median values and ranges may be sensitive to the sampling of models and studies in the database. Several recent studies have found model differences to be an important driver of scenario outcome variation[19,20], and have identified distinct 'model fingerprints'[21]. Yet, the impacts of the uneven representation of models and studies on findings presented in the IPCC reports have not yet been quantified. Although it is noted in Chapter 3 of the IPCC WGIII report that uncorrected database statistics may be misleading[11], such statistics are still used for headline AR6 findings.

We first analyse the influence of individual models and studies on database statistics that are used to report key findings in the AR6 WGIII Summary for Policymakers[22] (SPM). Secondly, we assess the impact of dominant models on median outcomes in the

CICERO Center for International Climate Research, Oslo, Norway. ✉e-mail: ida.sognnas@cicero.oslo.no

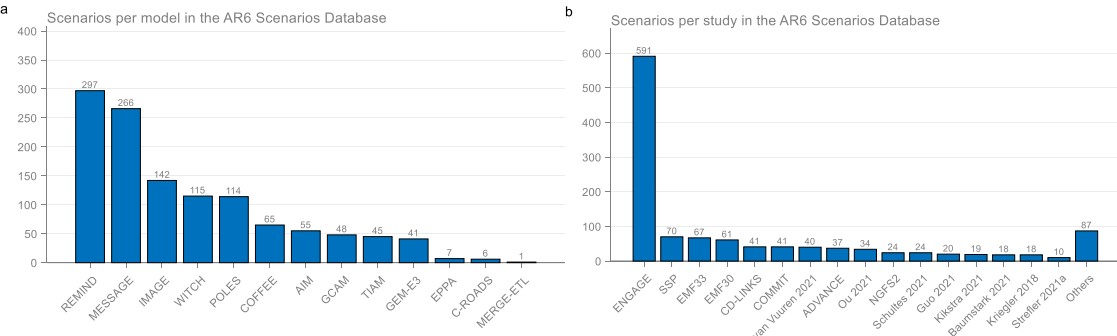

**Fig. 1 | Representation of models and studies in the IPCC AR6 Scenarios Database used to derive quantitative emissions scenarios findings. a** Number of scenarios by model. **b** Number of scenarios by study. All the scenarios that passed vetting and received a climate assessment are shown (1202 scenarios, out of 2304 submitted global scenarios[12]). Models are grouped into 'model families' representing potentially several versions of a single model core. The studies cover all scenarios in that study, though often from single papers. Scenario-based mitigation findings in the AR6 WGIII report are based on these scenarios (see "Methods"). Data: IPCC AR6 Scenarios Database[3].

AR6 scenarios database overall. Third, we compute model- and study-weighted medians and show that the results depend on the choice of weighting, in addition to the representation of models and scenarios. Finally, based on our findings, we discuss more fundamental issues with the use of database statistics to present emissions scenarios findings. Informed by the purpose of the IPCC and of emissions scenarios, we suggest ways forward for the IPCC authors to improve the assessment and communication of emissions scenarios findings.

## Results

### Influence of individual models and studies on key AR6 mitigation findings

To assess the influence of individual models and studies on key AR6 findings, we calculate the impact on findings reported in the WGIII SPM from removing each model and study one-by-one. WGIII SPM findings are reported using primarily median values, with either interquartile or 5-95th percentile ranges in parenthesis (see e.g., SPM Table SPM.2, or SPM paragraphs C.1.2 or C.3.2 in the AR6 WGIII report). Since the WGIII SPM is focused on scenarios that achieve 1.5 °C without overshoot (C1 category) and 2 °C (C3 category), we focus on 1.5 °C without overshoot and use 2 °C scenarios as a reference to assess the magnitude of the impacts. We analyse a total of 27 WGIII SPM findings (see Methods for the selection) and measure the impact of each individual model and study in two different ways: First, we measure the impact of removing the individual model and study on the 1.5 °C median relative to the difference between the 1.5 °C and 2 °C medians, and second, we measure how close the 1.5 °C median is to the individual model and study relative to the median of all the other models and studies. The first measure captures the impact of scenario sampling compared to the impact of switching from one climate target (1.5 °C) to another (2 °C). The second measure captures the influence of individual models and studies relative to all other models and studies within the same climate category (see Methods for details).

The impact on median values is substantial for several key AR6 findings (Fig. 2, Table 1). From removing just one model from the AR6 ensemble, median 2030 greenhouse gas (GHG) reductions in 1.5 °C scenarios (relative to 2019)—a widely recognised target[23] written into the 2022 Sharm el-Sheikh Implementation Plan[24]— shifts from 43% to 50%. Median 2030 $CO_2$ reductions shift from 48% to 56%. Median coal and gas reductions in 2050 shift from 95% to 83% and 43% to 29%, respectively. Net-zero GHG year shifts from 2098 to 2086 when removing one model, to 2084 when removing one study, and to after 2100 (the year of net-zero GHG is not specified in these scenarios) when removing several other models and studies.

For most of the assessed WGIII SPM findings, individual models have a much larger impact than individual studies (Table 1, Fig. 3 and Supplementary Figs. 1–3). One exception to this is the net-zero GHG year. For the net-zero GHG year, the ENGAGE study, with 25 of 97 scenarios in the C1 category, has a large influence because many ENGAGE scenarios do not reach net-zero GHG emissions before 2100 by design (Fig. 2 and Supplementary Note 1). The net-zero GHG year is also sensitive to the removal of models and studies because few scenarios reach net-zero around the median year (2098) (Fig. 4). The reason why median values are more sensitive to individual models is partly because the dominant model is responsible for a larger share of the C1 scenarios (41 of 97 scenarios) than the dominant study (25 of 97 scenarios). Models with fewer than 25 scenarios also have a smaller impact (Supplementary Figs. 1–3). In addition to this, models have a larger impact on median values because some WGIII SPM findings are more model-dependent than they are study-dependent. As seen in the cases of coal, oil, gas, $CH_4$, $N_2O$, and F-Gases, the level of disagreement across models is larger than the level of disagreement across studies (Supplementary Figs. 1–3). This is consistent with other studies that have found model differences to be larger than other scenarios differences[15,18,20,21].

The model with the largest impact on most median values is the model with the most scenarios, which, for all the assessed findings, is the REMIND model (Table 1). The strong influence on median values for this model is explained in large part due to its substantial share of the 1.5 °C scenarios in AR6 (41 of 97 scenarios). Because REMIND uses less coal and more oil compared to other models, median coal is higher and median oil is lower when REMIND is removed. And because near-term emissions reductions in REMIND are lower than in most other models, median GHG and $CO_2$ reductions in 2030 are higher when REMIND is not included. The model with the second largest impact after REMIND is the model with the second most scenarios, MESSAGE (20 of 97 scenarios). Except for the net-zero GHG year, however, the impact of removing MESSAGE is relatively small (Table 1). The magnitude of the shift when removing models depends on both the number of scenarios and the position of those scenarios relative to other scenarios (Fig. 4).

The study with the largest impact on median values is also the study with the most scenarios, ENGAGE[18] (Table 1). And the study with the largest influence after ENGAGE is also the study with the second most scenarios, van Vuuren 2021[25,26]. This study also happens to contain all the IMAGE scenarios in the C1 category, and removing it is therefore equivalent to removing the IMAGE model. The impact of van Vuuren 2021 on median values is, however, relatively small (Table 1).

Interquartile and 5-95th percentile ranges are also influenced by the dominant model and the dominant study, with the interquartile

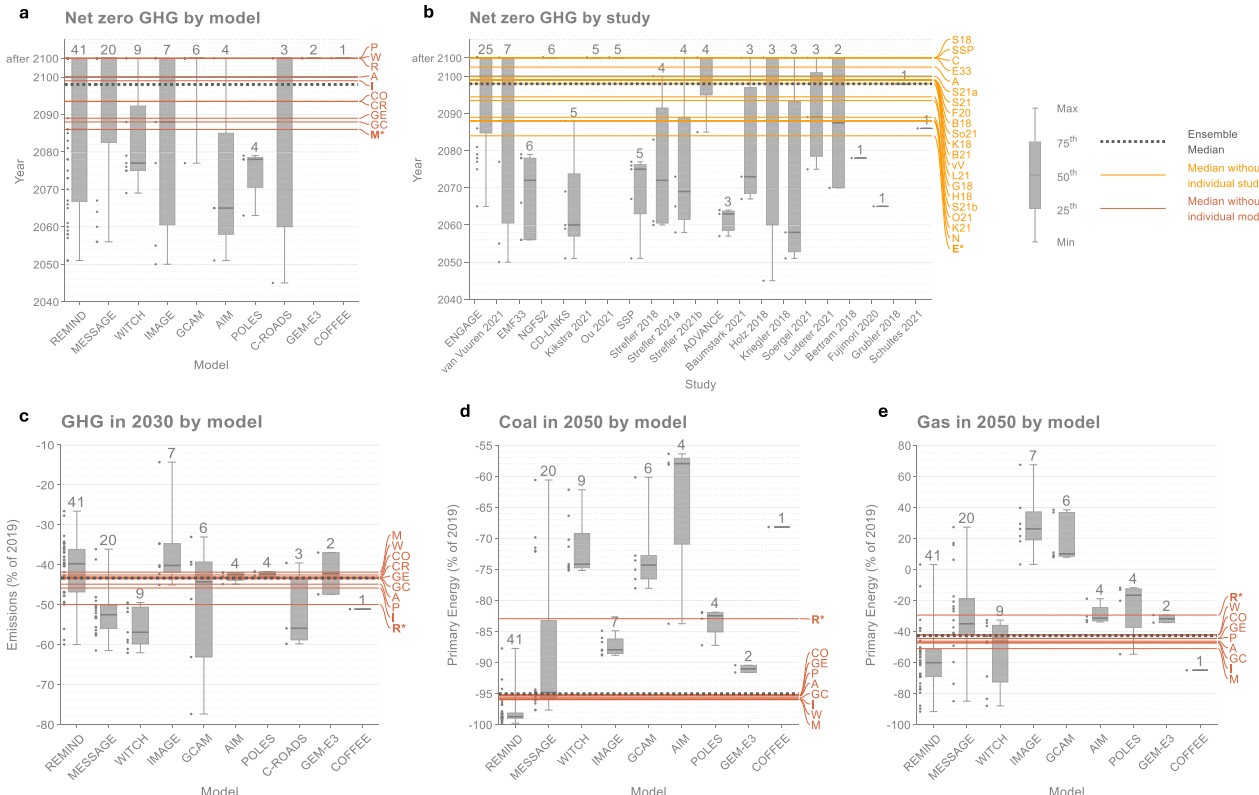

**Fig. 2 | Impact of removing individual models and studies on selected AR6 WGIII SPM findings.** Changes in median values for **a** Year of net zero GHG emissions by model, **b** Year of net zero GHG emissions by study, **c** GHG emissions in 2030 by model, **d** Coal in 2050 by model, and **e** Gas in 2050 by model. Boxes show the minimum and maximum, the interquartile ranges, and the median of each model/study. The number of scenarios from each model/study is shown at the top of each box, and the data points are shown to the left. Models and studies are ordered according to the number of scenarios, with the model/study with the most scenarios furthest to the left. Long, solid horizontal lines show median values when models (orange) and studies (yellow) are removed one-by-one, with letters at the end of each line indicating the model/study that has been removed. Bolded and starred letters show the model/study whose removal leads to the biggest shift in the median value. Dashed grey horizontal lines show ensemble medians. The findings

are selected based on a combination of policy relevance and impact (shown in Table 1) to illustrate how individual models and studies affect median values (more findings are shown in Supplementary Figs. 1–3). All variables are from scenarios that limit global warming to 1.5 °C (>50%) (C1 category). Values above 2100 on the y-axis indicate 'after 2100', which means zero was not reached before 2100 (and may not be reached). The model acronyms are R: REMIND, M: MESSAGE, W: WITCH, I: IMAGE, GC: GCAM, A: AIM, P: POLES, CR: C-ROADS, GE: GEM-E3, CO: COFFEE and the study acronyms are E: ENGAGE, vV: van Vuuren 2021, E33: EMF33, N: NGFS2, C: CD-LINKS, K21: Kikstra 2021, O21: Ou 2021, SSP: SSP, S18: Strefler 2018, S21a: Strefler 2021a, S21b: Strefler 2021b, A: ADVANCE, B21: Baumstark 2021, H18: Holz 2018, K18: Kriegler 2018, So21: Soergel 2021, L21: Luderer 2021, B18: Bertram 2018, F20: Fuji-mori 2020, G18: Grubler 2018, S21: Schultes 2021. Data: IPCC AR6 Scenarios Database[3].

ranges being more sensitive than the 5-95th percentile ranges, and with the dominant model having a larger impact than the dominant study (Fig. 3).

The median values that are least impacted by individual models and studies (relative to the differences in 1.5 °C and 2 °C medians) include peak $CO_2$ and GHG emissions years and net-zero $CO_2$ year (Table 1). Because almost all scenarios have the same peak GHG and $CO_2$ year (2020), removing individual models and studies has no impact. For the net-zero $CO_2$ year, inter-model variation is still considerable, but the two models with the most scenarios happen to both be very close to the ensemble median (Supplementary Table 1). Therefore, removing these two models has almost no impact on the median in this case. Models that give significantly different net-zero years (such as GCAM, POLES, C-ROADS, and COFFEE) do not have enough scenarios to influence the median net-zero year on their own. This highlights that it is not only the numbers of scenarios that matter, but also their positioning relative to the median.

Overall, the largest influence on the assessed WGIII SPM findings comes from the dominant model (Table 1). Compared to the dominant model, the dominant study and other models have much less influence (Supplementary Figs. 1–3). For 16 of the 27 SPM findings, the reported median is closer to the median of the dominant model than to the

median of all the other models combined (Table 1). For coal reductions, year of net-zero GHG emissions, $CH_4$ and F-gas emissions, the impact from the dominant model on the median is as large as or larger than the distance between reported median values for 1.5 °C and 2 °C scenarios. This shows that the uneven representation of models can be as or more important than the climate category for median scenario outcomes.

## Influence of dominant models on the AR6 scenarios database overall

We have so far focused on the scenario variables reported in the WGIII SPM. To assess the influence of dominant models on the AR6 scenarios database findings overall, which is relevant for users of the database more broadly[7,10,13,27,11], we compute the impact on the median values of all the variables in the database (see "Methods"). In each climate category, we compute the number of variables for which the median (in 2050) is closer to the median of the dominant model than to the median of all the other models combined. We do this for all scenario variables reported by at least two models.

Most median values in the AR6 database are closer to the median of the dominant model than to the median of the other models taken together (Fig. 5). In scenarios that limit global warming to 1.5 °C (C1 category), 79% of median values are closer to the median of the

**Table. 1 | Impact of removing individual models and studies on median values in the AR6 WGIII SPM**

| Scenario findings[a] | SPM Source | 1.5 °C[b] Median | Model remove[d] (1.5 °C) New Median | Model | Impact measures (%)[e] Within | Between | Study remove[d] (1.5 °C) New Median | Study | Impact measures (%)[e] Within | Between | 2 °C[c] Median |
|---|---|---|---|---|---|---|---|---|---|---|---|
| F-gases, 2050 (%) | C.1.2 | 88 | 79 | R | 99 | **207** | 88 / 88 | vV* / E | 50 / 1 | 5 / 1 | 83 |
| CH₄, 2050 (%) | C.1.2 | 51 | 59 | R | 66 | **182** | 49 | E | 22 | **47** | 46 |
| Net zero GHG (yr) | Table SPM.2 | 2098 | 2086 / >2100[f] | M* / R | 63[f] / 37[f] | **171[f]** / **100[f]** | 2084 | E | 67[f] | **200[f]** | >2100[f] |
| Coal, 2050 (%) | C.3.2 | 95 | 83 | R | 77 | **99** | 96 / 96 | vV* / E | 10 / 58 | 7 / 4 | 83 |
| N₂O, 2050 (%) | C.1.2 | 26 | 31 | R | 19 | **82** | 23 | E | 41 | **37** | 19 |
| CH₄, 2050 (%) | C.1.2 | 44 | 48 | R | 69 | **66** | 43 | E | 14 | 10 | 37 |
| Gas, 2050 (%) | C.3.2 | 43 | 29 | R | 43 | **50** | 48 / 42 | vV* / E | 7 / 69 | 18 / 3 | 16 |
| Oil, 2050 (%) | C.3.2 | 61 | 72 | R | 52 | **38** | 60 / 62 | K21* / E | 6 / 25 | 5 / 1 | 32 |
| Oil wo. CCS in 2050 (%) | C.3.2 | 61 | 71 | R | 51 | **36** | 57 / 61 | vV* / E | 34 / 17 | 17 / 1 | 34 |
| CO₂, 2030 (%) | C.1.2 | 48 | 56 | R | 65 | **31** | 46 | E | 23 | 7 | 22 |
| GHG, 2030 (GtCO₂-eq/yr) | Table SPM.2 | 31 | 28 | R | 69 | **30** | 30 / 32 | vV* / E | 42 / 13 | 8 / 4 | 44 |
| GHG, 2030 (%) | Table SPM.2 | 43 | 50 | R | 65 | **29** | 46 / 43 | vV* / E | 44 / 8 | 11 / 2 | 21 |
| Coal wo. CCS in 2050 (%) | C.3.2 | 98 | 97 | R | 84 | **29** | 99 / 98 | vV* / E | 9 / 12 | 3 / 1 | 93 |
| Transport-related CO₂, 2050 (%) | C.8.1 | 59 | 53 / 62 | M* / R | 64 / 26 | 19 / 11 | 57 / 58 | vV* / E | 9 / 8 | 6 / 0 | 29 |
| Cumulative net-negative CO₂ (GtCO₂)[g] | Table SPM.2 | −215 | −188 | R | 57 | 16 | −253 | E | 20 | **22** | −40 |
| Gas wo. CCS in 2050 (%) | C.3.2 | 68 | 64 | R | 41 | 15 | 66 / 68 | S21a* / E | 15 / 10 | 7 / 1 | 38 |
| Cumulative CO₂ (GtCO₂)[h] | Table SPM.2 | 512 | 461 | R | 60 | 14 | 519 / 515 | N* / E | 10 / 7 | 2 / 1 | 888 |
| GHG, 2040 (GtCO₂-eq/yr) | Table SPM.2 | 17 | 16 | R | 60 | 13 | 17 / 17 | S21a* / E | 21 / 69 | 3 / 3 | 29 |
| GHG, 2040 (%) | Table SPM.2 | 69 | 72 | R | 51 | 12 | 70 / 70 | S21a* / E | 22 / 49 | 4 / 3 | 46 |
| GHG, 2050 (%) | Table SPM.2 | 84 | 86 | R | 57 | 10 | 85 | E | 38 | 6 | 64 |
| GHG, 2050 (GtCO₂-eq/yr) | Table SPM.2 | 9 | 8 | R | 54 | 9 | 8 | E | 33 | 5 | 20 |
| CH₄, 2030 (%) | C.1.2 | 34 | 33 / 35 | M* / R | 43 / 46 | 9 / 8 | 33 | E | 41 | 9 | 19 |
| CO₂, 2040 (%) | C.1.2 | 80 | 82 | R | 33 | 6 | 79 / 80 | N* / E | 20 / 11 | 3 / 0 | 51 |
| Net zero CO₂ (yr) | Table SPM.2 | 2052 | 2051 / 2052 | M* / R | 100 / 33 | 5 / 3 | 2051 | E | 25 | 5 | 2071 |
| Cumulative CO₂ (GtCO₂)[i] | Table SPM.2 | 324 | 307 / 311 | M* / R | 76 / 47 | 4 / 3 | 259 | E | 52 | 14 | 797 |
| Peak CO₂ (yr) | Table SPM.2 | 2020 | 2020 | R | 0 | 0 | 2020 | E | 0 | 0 | 2020 |
| Peak GHG (yr) | Table SPM.2 | 2020 | 2020 | R | 0 | 0 | 2020 | E | 0 | 0 | 2020 |

The table is sorted according to the 'Between' measure for the models. Numbers are rounded to the nearest digit. Figure 3 shows the top 13 findings that are most impacted according to this measure for either models or studies (values > 21%, bolded). See the expanded version of this table (Supplementary Table 1) for median values of the individual models and studies that are removed.
Models acronyms are R REMIND, M MESSAGE. Study acronyms are E ENGAGE, vV van Vuuren 2021, K21 Kikstra 2021, S21a Strefler 2021a, N NGFS2.
[a]Percentages denote reductions from 2019 (negative numbers are increases). Coal, oil, and gas are in primary energy.
[b]1.5 °C (>50%) with no or limited overshoot (C1 category).
[c]2 °C (>67%) (C3 category).
[d]The columns show the median and impact measures when the individual model/study with the largest impact is removed. When the model/study with the largest impact is different from the model/study with the most scenarios, the impact from the latter is shown in the second row. Models and studies that differ from the dominant model and study are marked with an asterisk.
[e]The impacts of removing individual models and studies are measured in two different ways. 'Between' is a unitless measure of the change in median value relative to the difference between the 1.5 °C and 2 °C medians and 'Within' is a unitless measure of how close the reported median is to the median of the removed model/study versus the median of all the other models/studies. Higher values denote higher impact (see "Methods" for details).
[f]'>2100' means after 2100. Impact measures may be over- or underestimates because the year can be earlier or much later (the year 2105 is used to calculate the impact).
[g]Between the year of net zero CO₂ and 2100.
[h]Between 2020 and the year of net zero CO₂.
[i]Between 2020 and 2100.

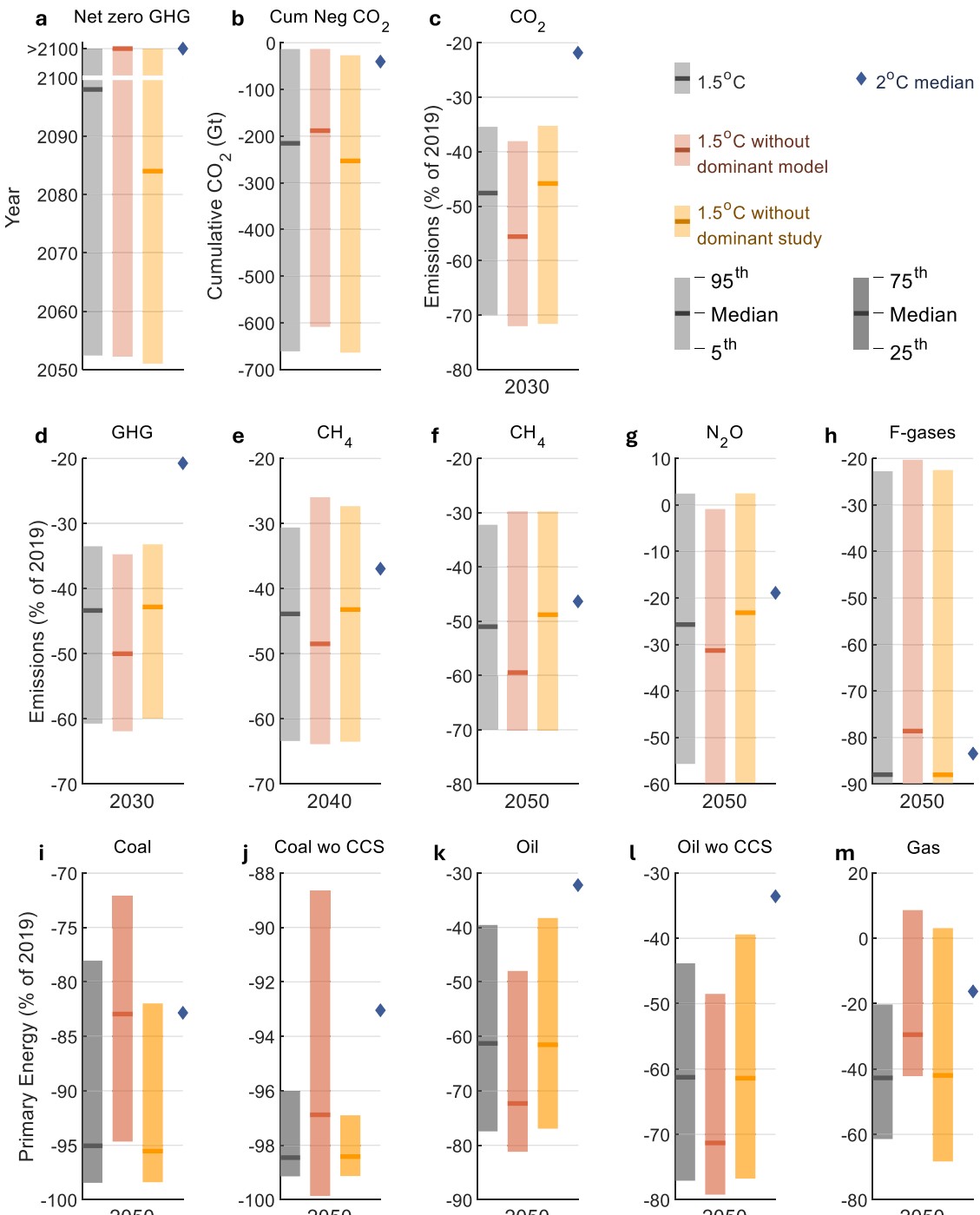

**Fig. 3 | Impact of removing the individual model and study with the most scenarios on selected AR6 WGIII SPM findings. a** Year of net zero GHG emissions, **b** Cumulative net-negative $CO_2$ emissions between the year of net zero and 2100, **c** $CO_2$ emissions in 2030, **d** GHG emissions in 2030, **e** $CH_4$ emissions in 2040, **f** $CH_4$ emissions in 2050, **g** $N_2O$ emissions in 2050, **h** F-gas emissions in 2050, **i** Coal in 2050, **j** Coal without CCS in 2050, **k** Oil in 2050, **l** Oil without CCS in 2050, **m** Gas in 2050. The selection represents the findings that are most impacted according to the 'Between' measure for either models or studies, as shown in Table 1. The different years and ranges correspond to what is reported in the SPM: For fossil variables, interquartile ranges are shown; for other variables, 5th-95th percentile ranges are shown. The grey bars show statistics for all scenarios that limit global warming to 1.5 °C (>50%) (category C1, including only vetted scenarios that received a climate assessment). The other bars show statistics when the model with the most scenarios is removed (orange bars) and when the study with the most scenarios is removed (yellow bars). The blue diamonds show median values of all scenarios that limit global warming to 2 °C (> 67%) (category C3, including only vetted scenarios that received a climate assessment). The model with the most scenarios is the REMIND model, and the study with the most scenarios is the ENGAGE study for all the findings. '>2100' means 'after 2100' (the year is not specified for scenario that reach net zero GHG emissions after 2100), 'CCS' stands for Carbon Capture and Storage. Data: IPCC AR6 Scenarios Database[3].

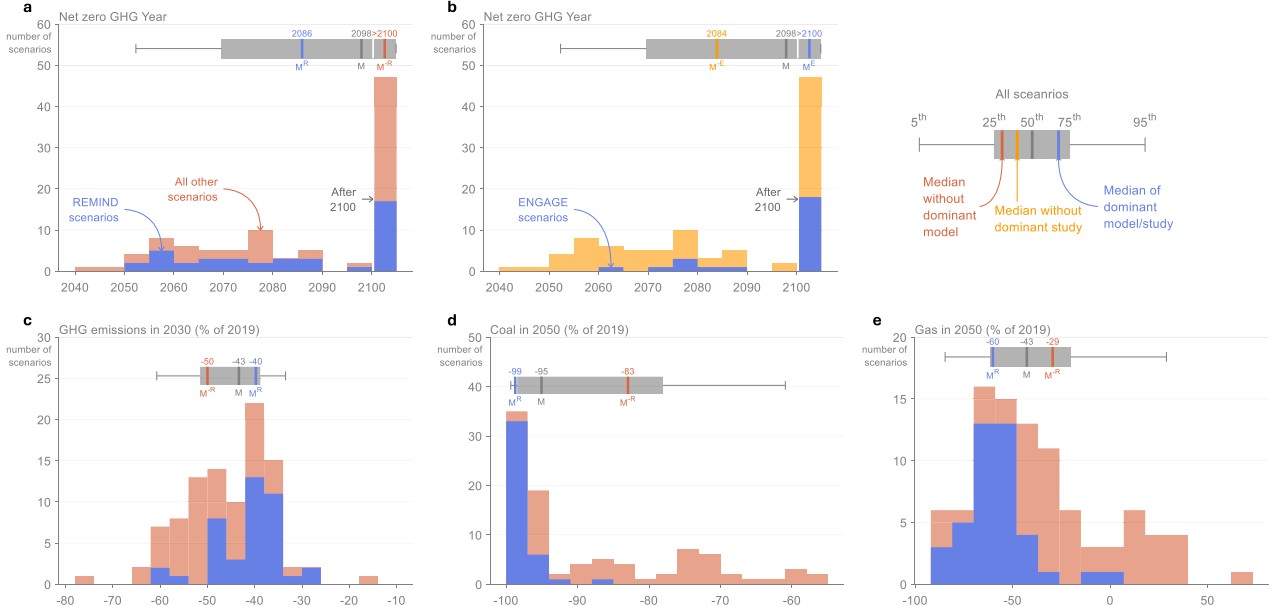

**Fig. 4 | Distribution of outcomes and impact on median values of removing the model and study with the most scenarios for selected AR6 WGIII SPM findings. a** Year of net zero GHG emissions by model, **b** Year of net zero GHG emissions by study, **c** GHG emissions in 2030 by model, **d** Coal in 2050 by model, and **e** Gas in 2050 by model. Dominant model/study scenarios are shown in blue, and all other scenarios are shown in orange/yellow. The findings are selected based on a combination of policy relevance and impact (shown in Table 1). All variables are from scenarios that limit global warming to 1.5 °C ( > 50%) (C1 category). The boxes at the top of each histogram show the 5th, 25th, 50th, 75th, and 95th percentiles of all scenarios (in grey), the median of the dominant model/study (in blue), and the median without the dominant model/study (in orange/yellow). '>2100' means after 2100 (net zero is not reached before 2100 and may not be reached in these scenarios). Data: IPCC AR6 Scenarios Database[3].

dominant model than to the median of all the other models. In scenarios that limit global warming to 2 °C (C3 category), 64% of median values are closer to the median of the dominant model than to the median of all the other models. Across all climate categories, 67% of median values in the AR6 scenarios database are closer to the median of the dominant model than to the median of all the other models.

Within each climate category, the dominant model depends on the scenario variable. This is because different models report different scenario variables (Supplementary Fig. 4 and Supplementary Data 1). In scenarios that limit global warming to 1.5 °C and 2 °C, the most common dominant model across variables is REMIND, followed by MESSAGE and IMAGE.

Different models dominate across variables in different climate categories (Fig. 5). Whereas REMIND is the most common dominant model in the C1, C3, C7, and C8 categories, MESSAGE is the most common dominant model in the C2, C4, C5, and C6 categories. This means that when comparing median outcomes of variables across climate categories—which is often done to show the implications of different climate targets—the differences may in some cases be more reflective of differences in model sampling than of the climate target. For example, comparing the implications of different levels of 'overshoot' (C1 versus C2) may be more about differences in the REMIND and MESSAGE models than about overshoot.

Overall, the models with the most scenarios in the AR6 database have a much larger impact on median values than the models with fewer scenarios. The two models with the most scenarios (47% of all vetted and climate-assessed scenarios), REMIND and MESSAGE, are responsible for 77% of the cases where the median is closer to the dominant model than to the median of all the other models. For 96% of the cases, the responsible model is one of the four models with the most scenarios in the AR6 scenarios database (REMIND, MESSAGE, IMAGE, and WITCH). The distribution of dominant models (Fig. 5b) is thus even more uneven than the distribution of scenarios per model (Fig. 1a).

The results are similar if we limit the analysis to include only Tier 1 and Tier 2 variables, which are considered the more important variables (see Methods and Supplementary Data 2). All models should submit Tier 1 variables (of which there are 82), suggesting that the uneven submissions across variables should be less common. In the C1 category, 63% of Tier 1 and 73% of Tier 2 variables have median values that are closer to the median of the dominant model than to the median of all the other models (Fig. 5). In the C3 category, 48% of Tier 1 and 59% of Tier 2 variables have median values that are closer to the median of the dominant model than to the median of all the other models. Across all climate categories, 47% of Tier 1 variables and 65% of Tier 2 variables have median values that are closer to the median of the dominant model than to the median of all the other models. In most cases, the dominant model is one of the four models with the most (vetted and climate-assessed) scenarios in the AR6 scenarios database. Models other than REMIND, MESSAGE, IMAGE and WITCH are the dominant model in less than 5% of all cases.

We expect models that report more variables (Supplementary Fig. 4) to be the dominant model for more scenario variables. However, models with more scenarios tend to also report more variables, which means that these two effects go in the same direction (Supplementary Fig. 5 and Supplementary Data 1). Overall, the number of vetted scenarios per model (Fig. 1 and Supplementary Fig. 6) is a better predictor of model dominance (Fig. 5) than the number of variables reported (Supplementary Fig. 4 and Supplementary Note 2).

**Median values depend on the choice of weighting and sampling**
The median values presented in the AR6 WGIII report are computed by giving each scenario equal weight. A simple way to counteract the uneven number of scenarios from different models and studies is to instead give each model or study equal weight (see Methods for details). For the assessed WGIII SPM findings, the direction of change in median values when each model or each study is given equal weight is usually the same as when the dominant model or study is removed

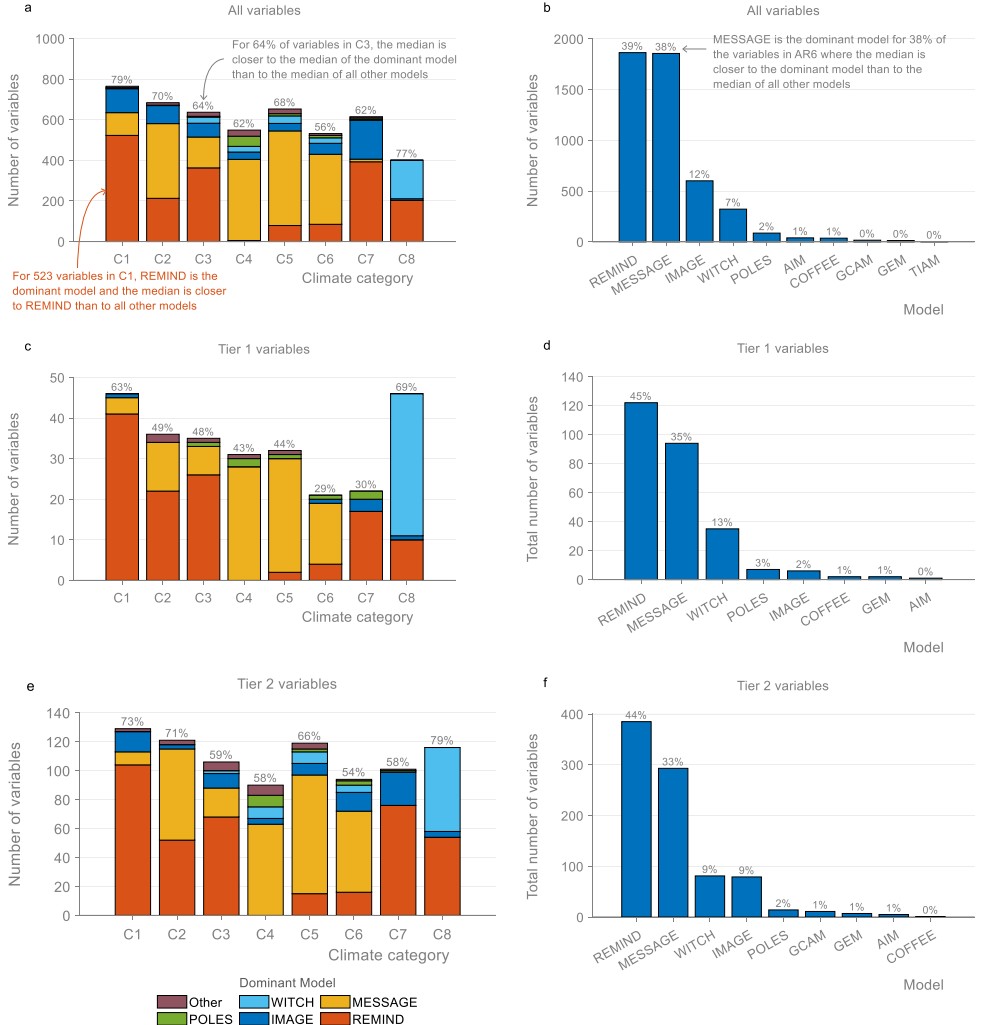

**Fig. 5 | Scenario variables with median values closer to the dominant model than to all other models.** The left column shows the number of variables with median values closer to the dominant model than to the median of all the other models, by climate category and by model (**a**, **c**, **e**). The right column shows the total number of variables across all climate categories with median values closer to the dominant model than to the median of all the other models, by model (**b**, **d**, **f**). Results are shown for all variables (**a**, **b**), Tier 1 variables (**c**, **d**), and Tier 2 variables (**e**, **f**) (see "Methods"). Percentages above bars in (**a**, **c**, **e**) show the proportion of variables in each climate category with median values closer to the dominant model than to the median of all other models. The colours show what model is the dominant model. Percentages above bars in (**b**, **d**, **f**) show the proportion of variables for which the median is closer to the dominant model than to all the other models, by the dominant model. Median values are in 2050. Only variables reported by at least two models in the AR6 Scenarios Database are included (see Methods). Data: IPCC AR6 Scenarios Database[3].

(Fig. 6). This is not surprising, given the scenarios from the dominant model and study are given much less weight in model- and study-weighted medians. When each model is given equal weight, median coal reductions in 2050 changes from 95% to 83% in 1.5 °C scenarios, which is the same as when removing the dominant model, and median gas reductions changes from 43% to 33%, which is close to the change from removing the dominant model (29%). Median GHG reductions in 2030 changes from 43% to 45%, which is less than when removing the dominant model (50%), because the models are evenly distributed on each side of the median for this variable.

Weighted medians, however, do not offer a straightforward solution to the problem of model and study representation. There are two reasons for this. First, model- and study-weighted medians often move in opposite directions, leading to large differences for certain key findings (Fig. 6 and Supplementary Figs. 7–9). The median net zero GHG year varies by almost two decades or more, depending on whether you give each scenario equal weight (2098), each study equal weight (2085), or each model equal weight (after 2100). This is partly

because the dominant model and the dominant study sit on opposite sides of the median. This shows that the choice of weighting scheme can be a key determinant of median values. But it is not clear what weighting scheme is more appropriate. Scenario outcomes can depend on model or study assumptions, or both, and it is not clear whether it is models or studies (or scenarios) that should be given equal weight in the calculation of database statistics. Second, model- and study-weighted medians may be dependent on what models and studies are included and not included in the AR6 scenarios database. While model- and study-weighted medians are insensitive to the number of scenarios, they are not insensitive to the representation of models and studies. Not all models submitted scenarios to the AR6 database, and of the more than 50 models that submitted scenarios, only 13 models submitted scenarios that passed vetting and received a climate assessment[11] (Supplementary Note 1 and Supplementary Table 2). Furthermore, because not all models report all variables in all climate categories, most findings are based on even fewer than those 13 models (Supplementary Fig. 10). Thus, the models that are used to

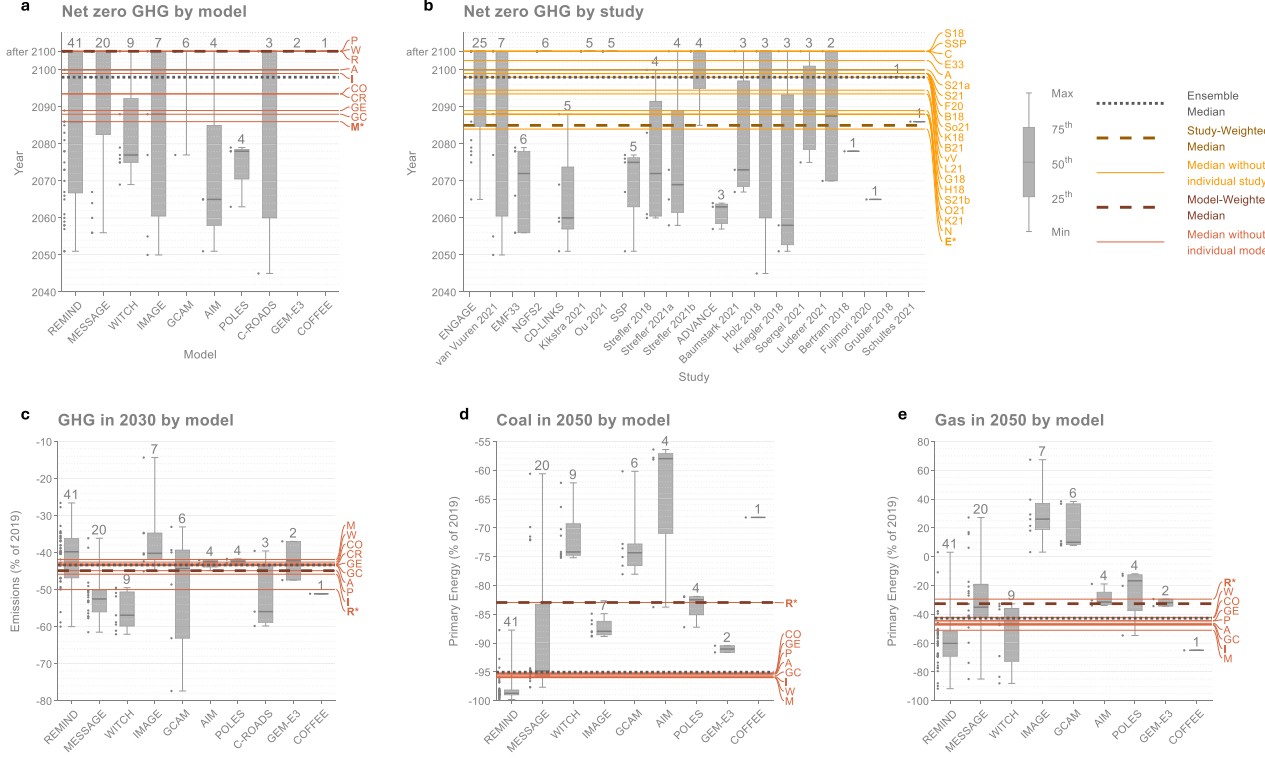

**Fig. 6 | Model- and study weighted medians for selected AR6 WGIII SPM findings. a** Year of net zero GHG emissions by model, **b** Year of net zero GHG emissions by study, **c** GHG emissions in 2030 by model, **d** Coal in 2050 by model, and **e** Gas in 2050 by model. Boxes show the minimum and maximum, the interquartile ranges, and the median of each model/study. The number of scenarios from each model/study is shown at the top of each box, and the data points are shown to the left. Models and studies are ordered according to the number of scenarios, with the model/study with the most scenarios furthest to the left. Long, solid horizontal lines show median values when models (orange) and studies (yellow) are removed one-by-one, with letters at the end of each line indicating the model/study that has been removed. Bolded and starred letters show the model/study whose removal leads to the biggest shift in the median value. Dashed grey horizontal lines show ensemble medians. Longer dashed horizontal lines show medians weighted by

model (orange) and by study (yellow). The findings are selected based on a combination of policy relevance and impact (shown in Table 1) to illustrate how individual models and studies affect median values (more findings are shown in Supplementary Figs. 1–3). All variables are from scenarios that limit global warming to 1.5 °C (>50%) (C1 category). Values above 2100 on the y-axis indicate 'after 2100', which means zero was not reached before 2100 (and may not be reached). The model acronyms are R: REMIND, M: MESSAGE, W: WITCH, I: IMAGE, GC: GCAM, A: AIM, P: POLES, CR: C-ROADS, GE: GEM-E3, CO: COFFEE and the study acronyms are E: ENGAGE, vV: van Vuuren 2021, E33: EMF33, N: NGFS2, C: CD-LINKS, K21: Kikstra 2021, O21: Ou 2021, SSP: SSP, S18: Strefler 2018, S21a: Strefler 2021a, S21b: Strefler 2021b, A: ADVANCE, B21: Baumstark 2021, H18: Holz 2018, K18: Kriegler 2018, So21: Soergel 2021, L21: Luderer 2021, B18: Bertram 2018, F20: Fujimori 2020, G18: Grubler 2018, S21: Schultes 2021. Data: IPCC AR6 Scenarios Database[3].

derive AR6 findings represent only a small subset of the models in the scenarios literature.

The problem of uneven sampling is not unique to the IPCC scenarios database. Unbalanced samples are common in social sciences, and several weighting methods have been developed to deal with this. These methods, however, rely on knowing the target population (see e.g.,[28,29].). When the target population is known, weights can be assigned to ensure that the distribution of chosen characteristics in the sample, such as age, gender, education, and geographical location, reflects the target population. For the IPCC scenarios ensemble, however, the target population is unclear. Should it be all plausible scenarios, including ones that have never been modelled, or should it be all scenarios that have been published? And what scenario characteristics should one be ensuring a good representation of? While median fossil fuel reductions depend mostly on model representation, the net-zero year depends on both model and study representation. This suggests that the relevant characteristics depend on the scenario outcome in question.

Weighting methods have also been discussed extensively in relation to physical climate modelling[30–33]. Most likely climate outcomes (contingent on forcing levels) are most often computed in structured experiments where each climate model is given equal weight[34]. In the literature, it has been argued that greater weight should be given to

climate models that have been shown to have greater skill and to models that are more independent[31,33]. That is, climate models should be weighed based on both performance and model independence, where dependencies may stem from the sharing of ideas for parametrization or simplifications, or from sharing of computer code, which may lead to similar model biases[31]. Because different climate models have greater skill at simulating different climate outcomes, and because different outcomes depend on different assumptions, both performance and independence depend on the outcome in question[33].

Despite valuable suggestions for how to evaluate IAMs[35] there are no agreed upon performance metrics for IAMs. While climate model performance is based on comparing model projections with historical observations[33], observations that can be used to assess the performance of IAMs are not available in the same way (Supplementary Note 3). And research into IAM dependencies is lacking.

In lieu of performance and independence metrics, weighting by model might represent an improvement to the current approach of weighting by scenario, as it removes duplication of model outcomes that may result from known model fingerprints. Giving each model equal weight is also the most common approach in climate modelling. But unlike climate model ensembles, where each model is run once for each scenario in a structured experiment[34] (such as in Coupled Model Intercomparison Projects (CMIPs)), the IPCC IAM ensembles contain a

diversity of different scenarios, run by different sets of models, under different assumptions, to answer different questions. In this case, differences in outcomes do not represent different answers to the same question, which may be interpreted as the uncertainty of the answer, but how the answers change when the questions change. In this case, weighting by model remains an arbitrary approach, which may also remove important variation captured by different studies run by the same model.

## Discussion

We have shown that several WGIII SPM median values are influenced considerably by the model, and in some cases study, with the most scenarios in the AR6 database. Additionally, we have shown that weighting does not offer a straightforward solution to the uneven representation of models and studies. The median will, in any case, depend on the weighting choice and the representation of models and studies. This brings up more fundamental questions regarding the use of database statistics to present emissions scenarios findings. Informed by the purpose of the IPCC assessment, we discuss three issues and make recommendations based on our findings and the kinds of insights that can be obtained from IAM scenarios.

First, median values and percentiles do not convey the level of agreement in findings, which is key to informing confidence and robustness. The IPCC is set up to "tell policymakers what we know and don't know" and "where there is agreement in the scientific community, where there are differences of opinion, and where further research is needed"[36]. As part of this, assessment findings with stronger agreement and multiple lines of evidence can be assigned a higher degree of confidence[37]. While interquartile and 5th-95th percentile ranges show the spread in scenario outcomes, they don't provide information about whether different models and studies agree or disagree. There is relatively low agreement, for example, on the precise reductions of emissions, coal, and gas in specific years, and on the net-zero GHG year. This is because these outcomes clearly depend on choices and assumptions that differ across models and studies (Figs. 2, 3 and 5). As our findings show, the reporting of descriptive statistics to the nearest percentage point for emissions reductions, the nearest 5% for coal and gas, and the closest five-year interval for the net-zero year is not robust to the sampling of models and studies. The reporting of median peak $CO_2$ and GHG emissions years is (Table 1). If the point is to show differences in implications between different climate targets, robust scenario findings can be defined as those that vary more across climate categories than they do across models and studies[20]. This robustness could be evaluated, for example, by assessing the sensitivity of outcomes to removing single models and studies or to giving models and studies equal weight, as is done in this paper.

Rather than focusing on median values and percentiles, which may not only be sensitive to sampling but can also be misinterpreted as probabilistic confidence intervals, the IPCC could report the full ranges of scenario outcomes and focus more on how outcomes depend on assumptions, including model and study assumptions. Many different strategies are consistent with 1.5 °C and 2 °C, and scenario analysis is largely about showing the different implications and trade-offs associated with different choices[38]. Scenario analysis is less suited to providing precise outcomes for specific variables[38]. In addition to this, several models and studies fall entirely outside of the interquartile, and in some cases, 5th-95th percentile, ranges that are used to present findings (Supplementary Figs. 1–3). But given the non-probabilistic nature of the emissions scenario ensemble[2] and the role of emissions scenarios in exploring different strategies and trade-offs, these results are no less important. An improved assessment of how scenario outcomes depend on choices and assumptions could be informed by several recent studies that have analysed how the model used, the scenario assumptions, and the climate target affect scenario outcomes[19–21]. A focus on such dependencies could provide policy-

relevant insights regarding, for example, how the net-zero emissions year depends on key assumptions, including the climate target definition, which is defined differently in different studies[18], and the discount rate, which may be defined differently in different models or studies[39].

Second, the reliance on database statistics to present key scenarios findings give a lot of weight to the subset of scenarios that are submitted to the database, pass vetting, and receive a climate assessment. This comes at the exclusion of findings that are captured by models and studies that are not included in this subset but still contribute to the literature. This is an issue because the IPCC is meant to assess the full scenarios literature, and the scenarios database is meant to aid this assessment, not replace it. With the construction and use of the scenarios database to derive mitigation findings, the IPCC is essentially conducting a meta-analysis of an "ensemble of opportunity"[2] that includes only a subset of the scenarios literature[12]. But the AR6 is meant to[40] "take all available literature on emissions scenarios fully into account independently of whether underlying emissions scenarios are submitted to the AR6 scenario database". To serve this goal, the IPCC should assess whether the subset of scenarios used to derive findings gives a good representation of the scenarios literature. Guidelines for the use of database statistics that address the issues of over- and underrepresentation could also be developed. Making this an integral part of the use of the IPCC scenarios database may help reduce biases in key findings that are based on the database.

Third, database statistics may not be very meaningful given the varied representation of different research questions and assumptions. According to AR6, "scenarios are neither predictions nor forecasts, but are used to provide a view of the implications of developments and actions"[2]. These views are provided through specific modelling studies that employ specific models and scenario assumptions to answer specific research questions, such as: What are the cost implications of meeting stringent climate targets without overshoot[18]? What are the impacts on electrification if renewable energy costs continue to decline?[41] How might changes in energy service provision affect global energy demand and supply, and the achievement of climate and development goals[42]? Although the scenarios database has an advantage in terms of the number of models and studies and therefore diversity of assumptions, this diversity also means that distributions of scenario outcomes are difficult to interpret[6] and that database statistics may not be very meaningful[2,27]. More readily available information on model and scenario assumptions[1,13], greater openness and transparency in the curation of the IPCC scenarios database including on vetting[12], and more research into the different causes of scenario outcome variability (e.g[17].) would enable users to get a better understanding of the representation of key assumptions in the database, which could help them make more meaningful comparisons and draw more relevant insights suited to their needs. But this information does not in itself make the scenarios in the database more comparable[43] or statistical values more meaningful. Modelling is for "insights not numbers"[44] and model outcomes carry less meaning when they are not interpreted with respect to the choices and assumptions under which they were generated, and the research questions they were designed to answer. Because of this, the insights that IAM scenarios offer are best understood and appreciated by assessing scenario papers directly.

The use of scenario database statistics to present key scenario findings, and giving each scenario equal weight, is an easy choice, but it is not a neutral choice as it gives more weight to choices and assumptions embedded in models and studies that have a large number of scenarios in the database. While the uneven representation of studies in the AR6 ensemble might mean that certain questions, and therefore certain answers, are overrepresented, the uneven representation of models might mean that certain mitigation strategies are overrepresented. Our results confirm that different models tend to

choose different strategies for reaching the same climate target[15,19–21] and thus provide different views of the implications of mitigation. REMIND 1.5 °C scenarios, for example, use less coal and gas, but more oil, and they mitigate more slowly in the beginning, but faster later on, compared to scenarios from other models.

More research is needed to understand what a representative sample of emissions scenarios might look like and how different models and studies contribute to this. Given that the IPCC scenarios database has a diversity of uses, there might not be a one-size-fits-all method for analysing it, and a transparent discussion of how to leverage it for different purposes, contexts and outputs is important. One way to remedy the uneven representation of models and studies in the database is to use the scenarios literature to better inform and guide the interpretation of different scenario outcomes. Even though the scenarios literature is also prone to an uneven representation of model and study assumptions, the weight given to different insights in the literature does not reflect a simple counting of scenarios and thus does not suffer in the same way.

The current use of database statistics to present key scenario findings in IPCC reports might mean that targets and decisions are based on findings that are more reflective of idiosyncratic model or study assumptions than of the chosen climate target and the scenarios literature. Moving away from descriptive statistics that are difficult to interpret and sensitive to sampling might be more in line with both the goal of the IPCC, to assess the full scenarios literature, and the purpose of integrated assessment modelling, to provide insights, not numbers.

## Methods

### AR6 scenarios analysed
The statistical values that are reported in the AR6 WGIII report and that are analysed in this paper are based on the subset of global scenarios in the AR6 scenarios database that passed vetting and that received a climate assessment (which means they were categorised into one of the C1-C8 climate categories). A total of 2304 global scenarios were submitted to the AR6 scenarios database. Of these, 1686 passed vetting and 1202 passed vetting and received a climate assessment[3].

The analysis of the influence of individual models and studies on key AR6 findings (Figs. 2, 3 and 5) includes scenarios only in the C1 and C3 climate categories, as these categories (1.5 °C with no or limited overshoot and 2 °C) are the focus in the WGIII SPM. It includes a select number of scenario variables based on what is reported in the WGIII SPM (see below). The analysis of the impact of dominant models on the AR6 scenarios database overall includes scenarios from all climate categories (C1-C8), and it includes all the scenario variables reported in the database (more detailed information on this is provided further below).

### Selection of AR6 WGIII SPM scenario variables
The AR6 WGIII SPM findings mainly cover scenarios that limit global warming to 1.5 °C (>50%) with no or limited overshoot (C1 climate category) and scenarios that limit global warming to 2 °C (>67%) (C3 climate category). We analyse 27 global 1.5 °C scenario (C1 category) variables whose median values are reported in the AR6 WGIII SPM. The list of all the 27 variables and the impacts of removing the models and studies with the largest impact and the models and studies with the most scenarios, is shown in Table 1.

### Impact measures
The impacts of removing individual models and studies are measured in two different ways, both shown in Table 1. 'Between' measures the impact relative to the differences between 1.5 °C and 2 °C scenarios: it is a unitless measure of the change in median value when an individual model/study is removed relative to the difference between the 1.5 °C and 2 °C medians. A value of 100% means that the change in median value is the same as the (absolute) difference between the 1.5 °C and

2 °C medians. A value of 0% means that there is no change.

$$Between = \frac{\left| Median^{1.5°C} - Median_{without\,X}^{1.5°C} \right|}{\left| Median^{1.5°C} - Median^{2°C} \right|} \times 100 \qquad (1)$$

where $X$ is an individual model or study. 'Within' measures the impact relative to other models and studies within the same (1.5 °C) climate category: it is a unitless measure of how close the reported median is to the median of the individual model/study that is removed versus the median of all the other models/studies combined. A value of 100% means that the median coincides with the median of the dominant model/study, and a value of 0% means that the ensemble median coincides with the median of all the other models/studies. A value above 50% means that the reported median is closer to the median of the dominant model/study than to the median of all the other models/ studies taken together.

$$Within = \frac{\left| Median^{1.5°C} - Median_{without\,X}^{1.5°C} \right|}{\left| Median_{of\,X}^{1.5°C} - Median_{without\,X}^{1.5°C} \right|} \times 100 \qquad (2)$$

where $X$ is an individual model or study.

### AR6 climate categories
We use the climate categories as defined in the AR6 WGIII report[2]. While the analysis of WGIII SPM findings includes scenarios only from the C1 and C3 categories, the analysis of the overall impact of dominant models on the AR6 scenarios database includes scenarios from all (C1–C8) categories.

Category C1 includes scenarios that limit warming to 1.5 °C in 2100 with a likelihood of greater than 50% and that reach or exceed warming of 1.5 °C during the 21st century with a likelihood of 67% or less. These scenarios are referred to in the AR6 report as scenarios that limit warming to 1.5 °C (>50%) with no or limited overshoot. Limited overshoot means exceeding 1.5 °C by up to about 0.1 °C for up to several decades. Category C3 includes scenarios that limit peak warming to 2 °C throughout the 21st century with a likelihood of greater than 67%. These scenarios are referred to in the AR6 report as scenarios that limit warming to 2 °C (>67%).

Category C2 includes scenarios that limit warming to 1.5 °C in 2100 with a likelihood of greater than 50% and exceed warming of 1.5 °C during the 21st century with a likelihood of greater than 67%. These scenarios are referred to in AR6 as scenarios that return warming to 1.5 °C (>50%) after a high overshoot. High overshoot refers to temporarily exceeding 1.5 °C global warming by 0.1 °C–0.3 °C for up to several decades. Categories C4, C5, C6 and C7 include scenarios that limit warming to 2 °C, 2.5 °C, 3 °C, 4 °C, respectively, throughout the 21st century with a likelihood of greater than 50%. Category C8 includes scenarios that exceed a warming of 4 °C during the 21st century with a likelihood of 50% or more.

### Grouping of models
To capture core model structure, we group what would generally be considered different versions of the same model (in model documentation and publications) together. For the scenarios that passed vetting and received a climate category, this results in 13 unique models. The list of unique model names and corresponding model versions is provided in Supplementary Table 3, together with the model acronyms used in tables and figures.

Due to differences in naming conventions across modelling groups, it is difficult to know how large the differences between different model versions are. Some modelling groups give their model a different name for each modelling study that the model takes part in (e.g., POLES), and other groups do not (e.g., AIM). Some modelling

groups use long and detailed model names to distinguish different versions and sub-versions, not only for the core model, but also for sub-models, from each other (e.g., MESSAGE and REMIND, linked to the land use models GLOBIOM and MagPIE).

Visual inspection of box plots (e.g., Figure 3 or Supplementary Figs. 1–3) suggests that the unique model names that we use capture core structural differences: there is a clear correlation between variable values and models.

## Grouping of studies
Study groupings are based on the 'Project_study' category in the AR6 scenarios database metadata[3]. 'Project_study' describes the parent project or individual study from which each scenario derives. This category is kept intact, as given in the metadata. This results in a total of 30 studies for the scenarios that passed vetting and received a climate category. The list of studies is provided in Supplementary Table 4, together with the acronyms used in tables and figures.

## Median values with individual models and studies removed
Scenarios are first assigned to models and studies from which they derive, according to Supplementary Tables 3, 4 (and as explained above). All the scenarios from each model/study are then removed from the ensemble, one-by-one, to calculate the median value without that model/study. The impacts from the models and studies with the largest impact are shown in Table 1, together with the impacts from the models and studies with the most scenarios (when these are different).

## Overall impact of dominant models on the AR6 database
The analysis of the overall impact of dominant models is complicated by the fact that i) each model reports different scenario variables (both from other models and across its own scenarios), and ii) each model has a different number of scenarios in each climate category. What model is the dominant model, therefore, depends on both the scenario category and the scenario variable: the same variable can be dominated by different models in different climate categories, and different variables within the same climate category can be dominated by different models.

To assess the overall impact of dominant models on the AR6 scenarios database, we compute the impact on each variable in each climate category of the model that (for that variable, in that climate category) has the most scenarios. More specifically, we compute the number of variables for which the median (in 2050) is closer to the median of the dominant model than to the median of all the other models combined, corresponding to a value above 50% for the 'Within' measure (Eq. (2) above). We do this for all scenario variables reported by at least two models, in each climate category. We exclude variables that are reported by only a single model because the median in this case is closer to the median of the dominant model than to the median of all other model scenarios (of which there are none) by default. Had we also included variables that are reported by only a single model, the percentage of median values that are closer to the median of the dominant model than to the median of all other models would be higher. If there is more than one model that has the most scenarios (for a given variable in a given climate category), we calculate the shift in median value of removing each one at a time and select the model whose removal leads to the greatest shift in the median to be the dominant model.

Because each climate category contains different scenarios from different models, and because different models report different variables (in different scenarios), the number of variables in each climate category differs (Supplementary Data 1). Only a subset of these, again, are reported by more than one model. There are 967 variables reported by at least two models in the C1 category, 978 variables in the C2 category, 1001 variables in the C3 category, 892 variables in the C4 category, 963 variables in the C5 category, 942 variables in the C6 category, 990

variables in the C7 category, and 518 variables in the C8 category. This amounts to a total of 7251 different median values, from at least two models, in the AR6 scenarios database.

## Tier 1 and Tier 2 variables
All global scenario data were submitted to the AR6 scenarios database using a common scenario reporting template. In the template, all variables are listed in a common format and assigned a Tier: "Tier 1 variables define a core set of information that would enable assessing the scenario in a meaningful way", "Tier 2 variables are important for enabling more specific analyses"[40], and remaining variables are not assigned a Tier. In case of constraints for providing scenario data, modelling teams were asked to consider the ranking of importance (Tier 1 and Tier 2).

Different Tiers were assigned by different chapters in WGIII. Of the 1871 variables that are listed in the common scenario reporting template, 303 are marked as 'core'. Of these core variables, 82 are Tier 1 and 221 are Tier 2. Of these, again, 79 Tier 1 variables and 201 Tier 2 variables are included in the AR6 scenarios database (Supplementary Data 2). These are the variables analysed in Fig. 5. The AR6 WGIII call for scenarios and templates is available at https://data.ene.iiasa.ac.at/ar6-scenario-submission/#/about.

## Weighted medians
The model- and study-weighted medians, shown in Fig. 6 and Supplementary Figs. 7–9, give each model and study equal weight. This is done by weighting each scenario from a model or study according to the inverse of the number of scenarios from that model or study.

## Data availability
All the data analysed is from the IPCC AR6 Scenarios Database, which is available through the AR6 Scenario Explorer and Database hosted by IIASA[3].

## Code availability
The code used for analysis in this study is available for download[45].

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

## Acknowledgements

I.S. and G.P.P. were supported by the European Union's Horizon Europe research and innovation programme under grant agreement no. 101056306 (IAM COMPACT) and no. 101081179 (DIAMOND), by the Norwegian Research Council, Finance Market Fund, project no. 309613 (Using scenarios to assess climate risks in the financial sector, StressTest), and the Norwegian Research Council project no. 334811 (Tracking Risks in Future Emission, Climate and Technological Assumptions, TRIFECTA).

## Author contributions

I.S. conceived of and designed the analysis and wrote the initial draft. I.S. and G.P.P. interpreted the data and wrote, read, and approved the final article.

## Competing interests

The authors declare no competing interests.
