## [Transparent Peer Review file · Nature Communications]

Influence of individual models and studies on quantitative mitigation findings in the IPCC Sixth Assessment Report

Corresponding Author: Dr Ida Sognnaes

Version 0:

Reviewer comments:

Reviewer #1

(Remarks to the Author)

The authors assess the impact of the uneven representation of models and modeling studies (mostly in terms of the underlying number of contributed scenarios) in the AR6 assessment by the IPCC concerning quantitative mitigation findings. The authors focus on changes to median values of mitigation findings when filtering out dominant models or studies from the assessed scenario ensemble. While the authors identify several key AR6 key findings, which are heavily influenced by respective dominant models/modeling studies, solutions to accounting for the uneven number of scenarios across models and underlying studies are not straightforward - three issues of using scenario ensembles to access findings are discussed, and suggestions for future improvements are made.

This assessment is very relevant due to the dominant role of median values and other descriptive statistics applied to scenario databases (which are often ensembles of opportunity rather than unbiased ensembles of the available scenario evidence) when informing climate policymakers. The aim and contribution of the manuscript are clearly defined and discussed. Overall, the manuscript is very well structured, and sufficient details on the methods and figures are provided to help readers understand and trace the author's approach. Therefore, I believe this is a valuable contribution to the literature. Before the manuscript is considered for publication, I have a few suggestions and clarification questions, which hopefully help to further tweak the manuscript.

I suggest expanding a little bit more on the "between" and "within" measures. You already describe the measures briefly on page 3 in lines 66-75 and in the Methods section under "Impact measures" - still, I believe a bit more reasoning for why these measures were chosen could be helpful, as to me, the measures were first not very intuitive. I suppose the idea of assessing median change, relative to the difference between climate categories C1-C2, is to illustrate that scenario filtering/weighting based on dominant models or studies can have a comparable impact on changes in mitigation benchmarks as switching to a different climate target. I believe readers would benefit from further motivating and describing why these measures were chosen.

Concerning the "within" measure, I was unsure whether there is a "x100" missing in the formula. I tried to reproduce some of the numbers of your impact measures in Table 1. However, as far as I can see, the median of the dominant model is not provided, which would be required to quickly check the "within" measure without having to consult the code. Maybe it is worth adding two columns to the table (one each for the "model remove" and "study remove"). For the "between" measure, I managed to reproduce some impact measures in Table 1 (e.g., for oil 2050: $(|61-72| / |61-32|) * 100 = 38$). However, for many, I cannot reproduce the impact measure based on the values provided in Table 1. I suppose this is because the values in Table 1 are rounded, correct? If so, maybe you could say that in a footnote to avoid confusion.

To facilitate the read, it would be good if you could revise the methods a bit to be clearer about when you only focus on C1 and C3, when about the SPM mitigation findings, and when you talk about all scenario categories in the AR6 Scenario Database. I started reading the Methods to get an idea of what you did, and I got a bit confused because it was not instantly clear to me whether you focus is on C1 and C3 or on all eight categories. In the section "AR6 scenario categories" you only refer to C1 and C3. Maybe this would be a good place to also introduce C2 and C4-8 and to briefly highlight when you are focusing on what in your analysis and why.

Page 4, lines 119-21: It may be worth highlighting that with "peak" you are referring to peak emissions and not peak concentrations.

Page 3, line 69: "and" instead of "ad"

Page 6, lines 160-162: "Thus, the distribution of dominant models (Figure 4) is more skewed than the distribution

of scenarios per model (Figure 1).” I am not sure I fully understand what you mean. Please consider rephrasing or clarifying this sentence.

(Remarks on code availability)

Reviewer #2

(Remarks to the Author)

This is an excellent manuscript that discusses the crucial issue of the use and interpretation of scenarios databases in IPCC reports, and especially the SPM. The use of statistical analysis in terms of median values to assess scenarios in a database that is recognised as an « ensemble of opportunities » is far from self-evident and definitely deserves more discussion. I found the manuscript did a great and necessary job of assessing the influence of individual models and studies on the IPCC findings regarding the mitigation scenarios. I also found the analysis and recommendations very balanced and carefully written.

Such an analysis would definitely have been relevant within the IPCC WG3 report itself.

I will not comment on the quantitative analysis since this is not my expertise. From a non-quantitative researcher point of view, it is very clearly explained and detailed, and the supplementary material is also very clear.

The focus of the article goes beyond the quantitative analysis, which it uses to outline the specificities of mitigation scenarios, and the importance of all the choices that are made in producing, weighing and analysing scenarios. It does not state that one method is inherently better than the others, but on the contrary shows the utmost importance of making the methods and their implications explicit. It also contributes to explaining some of the characteristics of the AR6 database-related finding by providing context about the ENGAGE study, and about the REMIND, MESSAGE and IMAGE models especially.

In a way, this manuscript is a re instantiation of Wynne and Keepin's analysis of the 1984 IASA Energy Study (<https://www.taylorfrancis.com/chapters/edit/10.4324/9780429282515-13/technical-analysis-iasa-energy-scenarios-bill-keepin-brian-wynne>), though much less confrontational, and I think we should have more of these studies that analyse model outputs together with their context of production and methodological choices.

Without naming it as such, the manuscript highlights the difficulty to maintain the boundary between research and assessment when working with these databases. One thing I get from it, and my own expertise on the organisation of work around databases both within IAM projects and IPCC reports, is that methods that make sense and are useful to the analysis of scenarios within a specific study (with its own clear purpose, protocols and questions, and targeted to people who are specialists and understand the context and limitations of the ensembles) are not necessarily suited to the purpose of an IPCC Assessment, which warrants the development of specific ways of analysing and presenting the data especially now that IPCC reports have a broader reach. It is an interesting evolution of a tool that was originally, in the 1990s and 2000s, developed to assess the literature and later became an instrument in research projects based on large model intercomparisons, requiring different design and methods.

So, in addition to the recommendations put forward in the article, perhaps one could be added along the lines of « recognising that scenarios databases have a diversity of uses and purpose, and that there should not be a one-size-fits-all method for analysing them, but rather a transparent discussion of how to leverage them for different purposes, contexts and outputs ».

I noted a couple of sentences that I did not fully understand, so consider rephrasing:

- lines 82-84

- line 218-222: it was not clear to me why this would apply to model- and study-weighted medians more than to scenario-weighted median

(Remarks on code availability)

Reviewer #3

(Remarks to the Author)

The manuscript highlights that within the IPCC database, models and studies are not equally represented even though the IPCC AR6 presents statistical ranges for different variables from the scenario database as high level findings. The authors show how these findings are not robust when one considers the uneven representation of models and scenarios within the database. I think this is an important issue to raise and the manuscript will represent a strong contribution to the literature examining scenarios and their role in the science-policy interface.

I have four main comments to strengthen the manuscript and increase its impact: (1) I think the main finding should be clearer and more salient; (2) I suggest that the authors better separate how they discuss the overrepresentation of models versus scenarios from particular studies; (3) I think that the authors are too dismissive of certain potential solutions for the

unbalanced nature of the IPCC database; (4) I think the authors should be more specific with regards to the issue of vetting and selection of certain insights into the IPCC process.

(1) Clarity and salience of findings.

The authors tackle the issue of overrepresentation of models and specific scenario studies by showing how the median values of the range change when you remove certain models and/or studies. Figure 3 shows this most clearly. (Though it's not clear why on Figure 3 different variables are shown for different years). I understand that on Figure 2 the authors want to show the impacts of particular models, but it's very hard to read. I suggest that the authors redesign this figure to make the message clearer. I also have trouble understanding the text from lines 149 to 188. The text discusses the difference in the median values of the sample from removing certain models and comparing that to the median value of all models which are left but at the moment it's very difficult to understand unless you're deeply embedded in dealing with these model ensembles.

(2) Separation of model versus study representation.

The results that the authors present mixes the discussion of the over representation of certain studies with the over representation of certain models. This is particularly the case in the section that starts online 60 in the section that starts online 189. I suggest that the authors tease these two apart and discuss them differently. As the authors' discussion section illustrates over sampling of certain models means something different scientifically than oversampling of different scenario designs and studies. This is because scenario designs and studies are designed to answer specific questions. In contrast models may have fingerprints but they tend to be used to ask the same questions. In other words it's not that message is designed to investigate one set of questions and remind another set. Therefore, I think it would strengthen the authors' argument to separate the discussion of the influence of the models and the influence of the representation of studies because the solution of the two is likely different (as the authors' discussion section suggests). This would also help strengthen the discussion of solutions (see next point).

(3) Discussion of solutions.

My second suggestion for discussing solutions is to more clearly delineate solutions related to model over representation and solutions related to study over representation because it's very likely that these two problems require different types of solutions.

With regards to study overrepresentation, the authors rightly know that in a way that IPCC scenario database miss uses the literature because it uses scenarios from different study protocols all in the same basket even though the scenario protocols were designed to answer different things. One potential solution for this would be to go back to the IPCC early scenario days and for the IPCC itself to commission scenarios with specific specifications. I'm not suggesting that this is the solution but the community itself has created this problem with how it treats scenarios and I think there needs to be a deeper reflection on how to fix the problem that the community itself has created.

With regards to model over representation, I suggest that the authors strengthen their discussion particularly around the treatment of unbalanced samples. Unbalanced samples are quite common in social science and I think there's something to be learned here from social science approaches to dealing with unbalanced samples. The authors present one potential solution of waiting medians but there are others and I think it would benefit the reader to have a more systematic overview of how unbalanced samples are managed and what might be appropriate to manage the unbalance sample of the IPCC scenario database.

Finally, with respect to the discussion of learning from unbalanced samples of climate models, I think that the authors are too dismissive here with a few potential avenues. For one the authors dismissed that it's possible to validate mitigation models with historical data. But I'm not sure why this is the case there are emerging approaches to use hindcasting to validate models, so why wouldn't we do that to deal with the issue of unbalanced samples? My second question is whether climate models also have fingerprints and if there's something to be learned there.

(4) Vetting.

I also suggest that the authors be more specific about the solutions to vetting. I fully understand the authors frustration with the current vetting approach in the IPCC scenario database and the overreliance of the scenario database for high-level findings and IPCC. I also have some sympathy for the IPCC authors who designed the designed the vetting protocol. Is there anything the authors can suggest to help resolve this impasse? I think the authors layout the problem very well and the insufficiency of current approaches about what should be done about it? One potential option is that the scenario database should be downplayed. Another option is that the scenario of database should be redesigned for example around research questions a third approach might be that the vetting procedure is just redesigned. These are all different options which imply different advantages and disadvantages and I think this article would be a very good place to highlight some of those.

(Remarks on code availability)

Reviewer comments:

Reviewer #1

(Remarks to the Author)

The authors have carefully addressed my previous comments. In my opinion, the paper is pretty much ready for publication. While reading through the revised manuscript I still noted two minor points (see below). These points can be easily addressed and I trust that this can be handled as part of the final paper editing.

- To me the text in the result section in lines 87-89 seems quite similar to the text in line 135-140 and therefore repetitive. Maybe this can be merged or some text could be cut here?
- In line 172, it should read "than of" rather than "as it is of".

(Remarks on code availability)

I did not review the Matlab code.

Reviewer #2

(Remarks to the Author)

My assessment of the original manuscript was already positive and it remains so with this version. My comments and suggestions were taken into account.

I have a few minor comments on some of the new text in revised sections.

- in the new discussion on unbalanced samples and comparison with climate model performance assessment, lines 257-260: I was a bit confused by the use of the "should", as I wasn't sure if the recommendations were from the cited references, or from the manuscript authors themselves. If the former, this could be clarified by adding "In the literature, it has been argued that..." or something to that effect. If the latter, this seems a bit strange given that we don't know what the authors would be basing these recommendations on.
- line 266: consider briefly explaining what "IAM dependencies" refer to - actually (I assume it refers to the same qualities), it may be helpful to define model independence/dependency in the previous paragraph already.
- line 326, "and the discount rate, which may be defined in the models": is a word missing? did you mean "defined differently"?

Just a comment on the evolution of databases, that does not imply any action on the authors' part:

line 337-339: this is somewhat ironic, given that the very first scenarios databases in the 1990s served as tools to assess whether IPCC reference scenarios covered the range in the literature !

(Remarks on code availability)

Reviewer #3

(Remarks to the Author)

The authors have addressed my concerns. Thank you for a good revision.

(Remarks on code availability)

I did not review the code.

REVIEWER COMMENTS

We would like to thank all the reviewers for the time taken and the many valuable and insightful comments, and positive assessment of the manuscript.

Please find our responses in blue.

Reviewer #1 (Remarks to the Author):

The authors assess the impact of the uneven representation of models and modeling studies (mostly in terms of the underlying number of contributed scenarios) in the AR6 assessment by the IPCC concerning quantitative mitigation findings. The authors focus on changes to median values of mitigation findings when filtering out dominant models or studies from the assessed scenario ensemble. While the authors identify several key AR6 key findings, which are heavily influenced by respective dominant models/modeling studies, solutions to accounting for the uneven number of scenarios across models and underlying studies are not straightforward - three issues of using scenario ensembles to access findings are discussed, and suggestions for future improvements are made.

This assessment is very relevant due to the dominant role of median values and other descriptive statistics applied to scenario databases (which are often ensembles of opportunity rather than unbiased ensembles of the available scenario evidence) when informing climate policymakers. The aim and contribution of the manuscript are clearly defined and discussed. Overall, the manuscript is very well structured, and sufficient details on the methods and figures are provided to help readers understand and trace the author's approach. Therefore, I believe this is a valuable contribution to the literature. Before the manuscript is considered for publication, I have a few suggestions and clarification questions, which hopefully help to further tweak the manuscript.

Thank you very much for the time taken to review our manuscript. We greatly appreciate your positive and constructive feedback, and we hope and believe that the changes have improved the manuscript.

I suggest expanding a little bit more on the “between” and “within” measures. You already describe the measures briefly on page 3 in lines 66-75 and in the Methods section under “Impact measures” - still, I believe a bit more reasoning for why these measures were chosen could be helpful, as to me, the measures were first not very intuitive. I suppose the idea of assessing median change, relative to the difference between climate categories C1-C2, is to illustrate that scenario filtering/weighting based on dominant models or studies can have a comparable impact on changes in mitigation

benchmarks as switching to a different climate target. I believe readers would benefit from further motivating and describing why these measures were chosen.

Thank you for this useful feedback. We have expanded the descriptions in the main text (lines 71-77), where the measures are introduced, and to the methods (lines 547-548 and 554-555), to make the intuition behind the measures, and their interpretation, clearer.

Concerning the “within” measure, I was unsure whether there is a “x100” missing in the formula. I tried to reproduce some of the numbers of your impact measures in Table 1. However, as far as I can see, the median of the dominant model is not provided, which would be required to quickly check the “within” measure without having to consult the code. Maybe it is worth adding two columns to the table (one each for the “model remove” and “study remove”). For the “between” measure, I managed to reproduce some impact measures in Table 1 (e.g., for oil 2050: $(|61-72| / |61-32|) * 100 = 38$). However, for many, I cannot reproduce the impact measure based on the values provided in Table 1. I suppose this is because the values in Table 1 are rounded, correct? If so, maybe you could say that in a footnote to avoid confusion.

You are right there was a “x100 missing”. Thank you for pointing this out, which we have now corrected. We have also:

- Added information that numbers are rounded to the nearest digit in the Table 1 caption.
- Added an expanded version of Table 1 that includes the median of the dominant model/study and the model/study with the largest impact to the SI (SI Table 1). It was not possible to include two more columns without either making the text very small (making it difficult to read) or having to use landscape layout (which does not integrate so well into the main text). We therefore chose the second option and put the expanded table in the SI.

To facilitate the read, it would be good if you could revise the methods a bit to be clearer about when you only focus on C1 and C3, when about the SPM mitigation findings, and when you talk about all scenario categories in the AR6 Scenario Database. I started reading the Methods to get an idea of what you did, and I got a bit confused because it was not instantly clear to me whether you focus is on C1 and C3 or on all eight categories. In the section “AR6 scenario categories” you only refer to C1 and C3. Maybe this would be a good place to also introduce C2 and C4-8 and to briefly highlight when you are focusing on what in your analysis and why.

Thank you for the feedback. We have added a new paragraph to the first section in the Methods, the ‘AR6 scenarios analysed’ section (lines 531-538), to clarify when different

scenario categories are analysed. We have also expanded on the 'AR6 climate categories' section (lines 698-700 and 709-717), as suggested. We hope this makes it clearer.

Page 4, lines 119-21: It may be worth highlighting that with "peak" you are referring to peak emissions and not peak concentrations.

We have added "emissions" to clarify.

Page 3, line 69: "and" instead of "ad"

Thank you, this has been corrected.

Page 6, lines 160-162: "Thus, the distribution of dominant models (Figure 4) is more skewed than the distribution of scenarios per model (Figure 1)." I am not sure I fully understand what you mean. Please consider rephrasing or clarifying this sentence.

We reworded and made the figure references more specific (including the figure letters) to make the meaning of this sentence clearer: "The distribution of dominant models (Figure 5b) is thus even more uneven than the distribution of scenarios per model (Figure 1a)"

Reviewer #2 (Remarks to the Author):

This is an excellent manuscript that discusses the crucial issue of the use and interpretation of scenarios databases in IPCC reports, and especially the SPM. The use of statistical analysis in terms of median values to assess scenarios in a database that is recognised as an « ensemble of opportunities » is far from self-evident and definitely deserves more discussion. I found the manuscript did a great and necessary job of assessing the influence of individual models and studies on the IPCC findings regarding the mitigation scenarios. I also found the analysis and recommendations very balanced and carefully written.

Such an analysis would definitely have been relevant within the IPCC WG3 report itself.

I will not comment on the quantitative analysis since this is not my expertise. From a non-quantitative researcher point of view, it is very clearly explained and detailed, and the supplementary material is also very clear.

The focus of the article goes beyond the quantitative analysis, which it uses to outline the specificities of mitigation scenarios, and the importance of all the choices that are made in producing, weighing and analysing scenarios. It does not state that one methods is inherently better than the others, but on the contrary shows the utmost importance of making the methods and their implications explicit. It also contributes to explaining some of the characteristics of the AR6 database-related finding by providing context about the ENGAGE study, and about the REMIND, MESSAGE and IMAGE models especially.

In a way, this manuscript is a reinstatement of Wynne and Keepin's analysis of the 1984 IIASA Energy Study

(<https://www.taylorfrancis.com/chapters/edit/10.4324/9780429282515-13/technical-analysis-iiasa-energy-scenarios-bill-keepin-brian-wynne>), though much less confrontational, and I think we should have more of these studies that analyse models outputs together with their context of production and methodological choices.

Without naming it as such, the manuscript highlights the difficulty to maintain the boundary between research and assessment when working with these databases. One thing I get from it, and my own expertise on the organisation of work around databases both within IAM projects and IPCC reports, is that methods that make sense and are useful to the analysis of scenarios within a specific study (with its own clear purpose, protocols and questions, and targeted to people who are specialists and understand the context and limitations of the ensembles) are not necessarily suited to the purpose of an IPCC Assessment, which warrants the development of specific ways of analysing and presenting the data especially now that IPCC reports have a broader reach. It is an interesting evolution of a tool that was originally, in the 1990s and 2000s, developed to assess the literature and later became an instrument in research projects based on large model intercomparisons, requiring different design and methods.

We thank the reviewer for their time taken to review the paper, and for their very positive and constructive feedback, as well as their interesting and relevant reflections on the broader issues that the paper seeks to address one part of. We greatly appreciate all of the suggestions made and we hope and believe the changes made have improved the manuscript further.

So, in addition to the recommendations put forward in the article, perhaps one could be added along the lines of « recognising that scenarios databases have a diversity of uses and purpose, and that there should not be a one-size-fits-all methods for analysing them, but rather a transparent discussion of how to leverage them for different purposes, contexts and outputs ».

This is a great suggestion, which we have incorporated more or less directly into the discussion (lines 379-382): “Given that the IPCC scenarios database has a diversity of uses, there might not be a one-size-fits-all method for analysing it, and a transparent discussion of how to leverage it for different purposes, contexts and outputs is important.”

I noted a couple of sentences that I did not fully understand, so consider rephrasing:
- lines 82-84

Thank you for the feedback, this sentence was a little unclear. We have rephrased the sentence to clarify.

- line 218-222: it was not clear to me why this would apply to model- and study-weighted medians more than to scenario-weighted median

Thank you for the comment. It is true that model and study representation also affects scenario-weighted medians, because it affects what scenarios are in the ensemble. We have adjusted the section (lines 231-238) to clarify.

Reviewer #3 (Remarks to the Author):

The manuscript highlights that within the IPCC database, models and studies are not equally represented even though the IPCC AR6 presents statistical ranges for different variables from the scenario database as high level findings. The authors show how these findings are not robust when one considers the uneven representation of models and scenarios within the database. I think this is an important issue to raise and the manuscript will represent a strong contribution to the literature examining scenarios and their role in the science-policy interface.

I have four main comments to strengthen the manuscript and increase its impact: (1) I think the main finding should be clearer and more salient; (2) I suggest that the authors better separate how they discuss the overrepresentation of models versus scenarios from particular studies; (3) I think that the authors are too dismissive of certain potential solutions for the unbalanced nature of the IPCC database; (4) I think the authors should be more specific with regards to the issue of vetting and selection of certain insights into the IPCC process.

We greatly appreciate the thorough evaluation of our paper. We especially thank the reviewer for their positive assessment of the paper and its contribution, and their highly constructive feedback, which we believe have helped improve the paper. Please find our responses and descriptions of changes made below.

(1) Clarity and salience of findings.

The authors tackle the issue of overrepresentation of models and specific scenario studies by showing how the median values of the range change when you remove certain models and/or studies. Figure 3 shows this most clearly. (Though it's not clear why on Figure 3 different variables are shown for different years. The different years correspond to the years for which the findings are reported in the SPM (coal, oil and gas values are reported for 2050, emissions for 2030 and 2050, and in some cases for 2040, etc.). This also corresponds to Table 1. We have added a clarification to the Figure 3 caption on this.). I understand that on Figure 2 the authors want to show the

impacts of particular models, but it's very hard to read. I suggest that the authors redesign this figure to make the message clearer. We agree there was quite a lot going on in Figure 2. We wanted to show how scenario outcomes depend on models and studies, how median values change when you remove individual models and studies, what model/study has the largest impact, and the impact relative to the ensemble median and the 2C median (reflecting our two impact measures in Table 1). To simplify Figure 2, we have removed the side symbols, which showed the largest impact from removing an individual model/study relative to the ensemble median and the 2C median. To show the median impact relative to the ensemble median we have instead added new figures (new Figure 4) that focus specifically on this. We hope these changes make Figure 2 easier to read and that the additional figure helps to further demonstrate the paper's findings. I also have trouble understanding the text from lines 149 to 188. The text discusses the difference in the median values of the sample from removing certain models and comparing that to the median value of all models which are left but at the moment it's very difficult to understand unless you're deeply embedded in dealing with these model ensembles.

The analysis of the overall impact is complicated by the fact that i) each model reports different scenario variables (both from other models and across own scenarios), and ii) each model has a different number of scenarios in each climate category. What model is the dominant model therefore depends on both the scenario category and the scenario variable. The scenario variable 'Primary Energy|Coal', for example, is dominated by different models in different climate categories. In the C1 category, 'Primary Energy|Coal' is dominated by REMIND: 41 of the 94 C1 scenarios that report 'Primary Energy|Coal' comes from REMIND. In the C4 category, 'Primary Energy|Coal' is dominated by MESSAGE: 39 of the 159 C4 scenarios that report 'Primary Energy|Coal' come from MESSAGE (and only 16 come from REMIND). In both cases, the dominant model is the same as the model with the most scenarios in the climate category overall: REMIND has the most C1 scenarios (41 of 97) and MESSAGE has the most C4 scenarios (39 of 159). This is because 'Primary Energy|Coal' is reported by most models in most scenarios. But many scenario variables are reported by only some models. In these cases, the dominant model can differ from the model that has the most scenarios in the respective climate category. 'Final Energy|Solar', for example, is reported by only 5 REMIND scenarios, but 15 MESSAGE scenarios in the C1 category. Hence, MESSAGE is the dominant model for 'Final Energy|Solar' in the C1 category, despite REMIND having more C1 scenarios overall. (MESSAGE is also the dominant model for 'Final Energy|Solar' in the C4 category, where 39 MESSAGE scenarios and only 2 REMIND scenarios report this variable).

What we do in the section referred to (previously lines 149-188, now lines 147-206), is go through each variable, in each climate category, and see what the dominant model is

and whether the median value is closer to the dominant than to the median of all the other models (for this variable, in this climate category).

To clarify and better explain the analysis and the associated findings, we have made a number of changes:

- We have thoroughly edited and expanded the relevant section (lines 147-206).
- We have added explanations to the relevant figure (previous Figure 4, now Figure 5).
- We have expanded the associated Methods section (lines 615-647).

(2) Separation of model versus study representation.

The results that the authors present mixes the discussion of the over representation of certain studies with the over representation of certain models. This is particularly the case in the section that starts online 60 in the section that starts online 189. I suggest that the authors tease these two apart and discuss them differently. As the authors' discussion section illustrates over sampling of certain models means something different scientifically than oversampling of different scenario designs and studies. This is because scenario designs and studies are designed to answer specific questions. In contrast models may have fingerprints but they tend to be used to ask the same questions. In other words it's not that message is designed to investigate one set of questions and remind another set. Therefore, I think it would strengthen the authors' argument to separate the discussion of the influence of the models and the influence of the representation of studies because the solution of the two is likely different (as the authors' discussion section suggests). This would also help strengthen the discussion of solutions (see next point).

We thank the reviewer for these helpful reflections and suggestions. To better reflect the distinctions and to improve the arguments related to study and model impacts we have made several changes both to the discussions section (see next point) and to the sections referred to here. More specifically, we have:

- In the section on SPM findings, which started on line 60 (now line 61), we have: Brought the paragraph that discusses why the impact from individual models is larger than the impact from individual studies higher up such that this is now presented as the first main finding. This is now followed by two paragraphs that, separately, discuss the impacts of dominant models and dominant studies. Since one of our main findings is that the dominant model has a much larger impact than the dominant study (and than all other models and studies) – which comes from comparing the impacts from models and studies – we still find it meaningful to mix the presentation of the impacts of models and studies somewhat in this

section. This finding is what motivates the singular focus on dominant models in the analysis of the overall impact in the following section. We have reworded the last paragraph (lines 135-146) to make this clearer.

- In the section on weighted medians, which started on line 189 (now line 206), we have: Used the new and expanded discussion on unbalanced samples in social sciences (lines 242-253) and climate model weighting (lines 254-278) to also discuss issues of model versus study representation, and the challenges associated with giving each model equal weight in IAM ensembles, which unlike climate model ensembles mixes many different scenarios, in more detail. We hope the changes help deepen the understanding of the challenges, and thereby also strengthen the discussion of solutions. Although we have tried to more clearly separate between model and study impacts, we nonetheless believe that the two impacts are best understood as different parts of the same problem – which is that it is the underlying representation of different choices and assumptions, some of which are defined in the study and some of which are defined in the model, that determine the distribution of outcomes, and hence that the solution depends on the outcome in question. We also elaborate on this further in the revised discussion (see next point). A fundamental problem for IAM ensembles is that we do not know what a representative sample of models or studies (or really, underlying choices and assumption) would be.

(3) Discussion of solutions.

My second suggestion for discussing solutions is to more clearly delineate solutions related to model over representation and solutions related to study over representation because it's very likely that these two problems require different types of solutions.

To take into account the thoughtful reflections and suggestions offered by the reviewer in this and the previous point, we have substantially edited the Discussions section, as well as the Weighting section.

First, we have added an explicit discussion of representation in the new section on unbalanced samples in social sciences (lines 242-253). For IAM ensembles, a key problem is that the target population is unknown and that the characteristics that should be well represented have not been defined. We note that the scenario characteristics that matter will depend on the scenario outcome in question, and how fossil fuel reductions, for example, depend mostly on model assumptions, while net-zero year, for example, depend on both model and study assumptions.

Given this, we find that the solutions to model and study overrepresentation are not so easy to separate. As an example, constraints on technologies, may be implemented

either in the models themselves (and thus vary across models in a given study), or in a study protocol (and thus be the same across models in a given study, but markedly different from other studies).

Rather than offering separate solutions to model and study overrepresentation, which is challenging given the “correct” representation of models and studies is undefined, one of our main recommendations is that the assessment focuses more on how scenario outcomes depend on assumptions (whether these are defined via the models or the studies). As explained in the revised Discussion “A focus on such dependencies could provide policy relevant insights regarding, for example, how the net zero emissions year depends on key assumptions including the climate target definition, which is defined differently in different studies, and the discount rate, which may be defined in the models.”

We nonetheless agree that it is important to clearly reflect upon and discuss, and where relevant distinguish between, model and study impacts on scenario outcomes. We have thoroughly edited and significantly expanded both the Discussion and weighting section with this in mind, and we hope that this has led to a more informative analysis of both the problem and potential solutions.

More specific changes are described below.

With regards to study overrepresentation, the authors rightly know that in a way that IPCC scenario database miss uses the literature because it uses scenarios from different study protocols all in the same basket even though the scenario protocols were designed to answer different things. One potential solution for this would be to go back to the IPCC early scenario days and for the IPCC itself to commission scenarios with specific specifications. I'm not suggesting that this is the solution but the community itself has created this problem with how it treats scenarios and I think there needs to be a deeper reflection on how to fix the problem that the community itself has created.

We fully agree and we hope that our paper will provide a valuable contribution to a better understanding of the issues and challenges associated with the use of the scenarios database to provide mitigation findings, and that this again will contribute to a deeper and more thorough discussion of how emissions scenarios should be assessed and findings synthesized. In the revisions we have added and/or expanded on several points that we hope will contribute to an even deeper understanding of the issues and thus also the potential solutions:

- From approaches to unbalanced samples in social sciences (lines 242-253) we learn that it is unclear how to assign weights when the target population, and the chosen characteristics, are unclear.

- Note that this relates to both study and model representation, however, as assumptions in both models and studies may affect different scenario outcomes differently. This is closely related to our first recommendation in the Discussions (lines 289-327), to focus more of the assessment on how scenario outcomes depend on key assumptions. We have tried to clarify the role of model and study assumptions in this further by adding that, “while fossil fuel reductions are more model dependent, net-zero GHG year depends on both model and study assumptions”. Rather than recommending one solution for study overrepresentation and one solution for model representation, our recommendation is therefore that the IPCC focuses more of the assessment on how scenario outcomes depend on assumptions, including both model and study assumptions. Several recent studies have investigated these relationships. An improved assessment of such dependencies, we further explain, will provide more policy relevant insights regarding (for example, how the net zero year depends both on the climate target definition – which is often defined in the study protocol – and other assumptions, which may be defined in the models). This would also be more in line with the purpose of IAM scenarios, to show implications and trade-offs.
- In the new and expanded section on climate model weighting (lines 254-278) we now also note key differences between climate model ensembles and IAM ensembles, in particular how the latter do not represent structured experiments, but many different experiments. We explain how, in IAM ensembles, variation in outputs do not represent the uncertainty of the answer, but how answers change when questions change. And that weighing each model evenly might be an improvement to the current situation, as it removes potential duplication stemming from distinct model fingerprints, but a highly imperfect one because it may hide important study variation (lines 270-278)
- We have also edited the Discussions section throughout with an eye to more clearly relate our recommendations to model and study representation. In our third recommendation we have clarified that (lines 370-373) *“While the uneven representation of studies in the AR6 ensemble might mean that certain questions, and therefore certain answers, are overrepresented, the uneven representation of models might mean that certain strategies are overrepresented”*. We have further emphasized how different scenarios represent different “views of the implications”, and how the scenarios literature is a key resource for fully understanding the insights that IAM scenarios offer. And we have added that, *“Even though the scenarios literature is also prone to an uneven representation of model and study assumptions, the weight given to different insights in the literature does not reflect a simple counting of scenarios and thus does not suffer in the same way.”* Finally, we have also added a section

that reflects on recent developments and suggested improvements (lines 355-362).

With regards to model over representation, I suggest that the authors strengthen their discussion particularly around the treatment of unbalanced samples. Unbalanced samples are quite common in social science and I think there's something to be learned here from social science approaches to dealing with unbalanced samples. The authors present one potential solution of waiting medians but there are others and I think it would benefit the reader to have a more systematic overview of how unbalanced samples are managed and what might be appropriate to manage the unbalance sample of the IPCC scenario database.

We thank the reviewer for the suggestion to look at how unbalanced samples are dealt with in the social science. We use this in a new paragraph (lines 242-253) to provide a deeper understanding of the underlying challenges associated with representation in IAM ensembles. More specifically we learn that knowing the target population, and what characteristics to focus on, is central to the approaches that are used to correct unbalanced samples in social sciences. For the IPCC scenarios ensemble, both the target population and the key characteristics are unclear.

These insights are relevant to our second recommendation in the Discussions section (lines 327-342), which we have reworded and expanded in parts to make this link clearer. We have also added a concrete suggestion that the IPCC develop guidelines for the use of database statistics that address the issues of over- and underrepresentation (applying to models and/or studies) (lines 339-342).

Finally, with respect to the discussion of learning from unbalanced samples of climate models, I think that the authors are too dismissive here with a few potential avenues. For one the authors dismissed that it's possible to validate mitigation models with historical data. But I'm not sure why this is the case there are emerging approaches to use hindcasting to validate models, so why wouldn't we do that to deal with the issue of unbalanced samples? My second question is whether climate models also have fingerprints and if there's something to be learned there.

- We agree that much could be learnt from physical climate modelling and a far more comprehensive comparison between IAMs and climate models. While we cannot capture all of that in this paper, which is focused on quantifying the impacts of individual models and studies on AR6 mitigation findings, we have significantly expanded the discussion of weighing methods from physical climate modelling to provide more insights regarding the kinds of considerations that need to be made when assigning weights. Specifically, we discuss issues associated with differences in model performance, which is intimately related to historical data validation, model independence, and differences in the set up

between climate model and IAM ensembles (structured experiments vs. combination of many different studies) (lines 254-278).

- To address the question regarding model validation and the use of historical data, we have added a new Supplementary Note 3 on 'Challenges in the use of historical data to validate IAMs' where we explain in more detail why IAMs cannot currently be validated using historical data. We refer to this in the main text in the discussion of model performance weights.
- When it comes to fingerprints for climate models, the term 'fingerprints' here usually refers to particular climate change fingerprints in climate projections, and comparisons of these with observational data, for example in detection and attribution studies. Such fingerprints are related to model performance (which is based on comparisons with observational data, and is outcome dependent) and model dependencies (which have to do with whether or not climate models give similar outputs). Since the literature on climate model weighting focuses on performance and independence, rather than fingerprints, this is also what we chose to focus on in the new and expanded discussion on physical climate models.

(4) Vetting.

I also suggest that the authors be more specific about the solutions to vetting. I fully understand the authors frustration with the current vetting approach in the IPCC scenario database and the overreliance of the scenario database for high-level findings and IPCC. I also have some sympathy for the IPCC authors who designed the designed the vetting protocol. Is there anything the authors can suggest to help resolve this impasse? I think the authors layout the problem very well and the insufficiency of current approaches about what should be done about it? One potential option is that the scenario database should be downplayed. Another option is that the scenario of database should be redesigned for example around research questions a third approach might be that the vetting procedure is just redesigned. These are all different options which imply different advantages and disadvantages and I think this article would be a very good place to highlight some of those.

We fully agree with the reviewer that the question of vetting procedures and options is an interesting and important one. And we also see the need for vetting in many circumstances. For all these reasons we are currently working on a separate paper focused on vetting alone, where we try to address these issues. This paper includes detailed quantitative analyses of vetting impacts and goes into far more depth when it comes to possible solutions, including advantages and disadvantages.

In the current paper, our main point is that individual models and studies have a large impact on certain key mitigation findings and given that the IPCC is meant to assess the full scenarios literature, we recommend that the IPCC assesses whether the subset of scenarios used to derive key findings gives a good representation of this literature. Vetting thus plays a role in the current paper because it represents part of the sampling process (along with the voluntary submission by different modelling teams). We don't currently know whether the vetting, or the submission process, introduces biases, but we argue that this should be assessed, and that this assessment should be an integral part of the use of the IPCC scenarios database (lines 327-342). We have reworded this (second) recommendation slightly to hopefully make it clearer how the issue here relates to representation of the scenarios literature in the sample of scenarios that is assessed by the IPCC.

RESPONSES TO REVIEWERS' COMMENTS

We would like to thank all the reviewers for their time and effort taken in reviewing this manuscript. We greatly appreciate the positive and constructive feedback.

Please find our responses in blue.

Reviewer #1 (Remarks to the Author):

The authors have carefully addressed my previous comments. In my opinion, the paper is pretty much ready for publication. While reading through the revised manuscript I still noted two minor points (see below). These points can be easily addressed and I trust that this can be handled as part of the final paper editing.

Thank you very much for the positive assessment. We have addressed both points (see below).

- To me the text in the result section in lines 87-89 seems quite similar to the text in line 135-140 and therefore repetitive. Maybe this can be merged or some text could be cut here?

Thank you for pointing this out. We have cut, edited, and merged parts of the text to make it less repetitive.

- In line 172, it should read “than of” rather than “as it is of”.

Thank you for noticing. This has been changed as suggested.

Reviewer #1 (Remarks on code availability):

I did not review the Matlab code.

Reviewer #2 (Remarks to the Author):

My assessment of the original manuscript was already positive and it remains so with this version.

My comments and suggestions were taken into account.

I have a few minor comments on some of the new text in revised sections.

Thank you very much for the positive assessment and constructive feedback on the new text. We have made the changes as suggested (see below).

- in the new discussion on unbalanced samples and comparison with climate model performance assessment, lines 257-260: I was a bit confused by the use of the "should", as I wasn't sure if the recommendations were from the cited references, or from the manuscript authors themselves. If the former, this could be clarified by adding "In the literature, it has been argued that..." or something to that effect. If the latter, this seems a bit strange given that we don't know what the authors would be basing these recommendations on.

We have changed this to "In the literature, it has been argued that greater weight should be given to climate models that have been shown to have greater skill and to models that are more independent^{33,35}."

- line 266: consider briefly explaining what "IAM dependencies" refer to - actually (I assume it refers to the same qualities), it may be helpful to define model independence/dependency in the previous paragraph already.

Thank you for pointing this out. As suggested, we now explain what model independency/dependency means in the previous paragraph (lines 259-261). This now reads "...climate models should be weighed based on both performance and model independence, where dependencies may stem from the sharing of ideas for parametrization or simplifications, or from sharing of computer code, which may lead to similar model biases³³."

- line 326, "and the discount rate, which may be defined in the models": is a word missing? did you mean "defined differently"?

We have changed the sentence to: "...and the discount rate, which may be defined differently in different models or studies⁴¹."

Just a comment on the evolution of databases, that does not imply any action on the authors' part:

line 337-339: this is somewhat ironic, given that the very first scenarios databases in the 1990s served as tools to assess whether IPCC reference scenarios covered the range in the literature !

That is interesting! Thanks for sharing.

Reviewer #3 (Remarks to the Author):

The authors have addressed my concerns. Thank you for a good revision.

Thank you very much for the positive assessment!

Reviewer #3 (Remarks on code availability):

I did not review the code.